# Endophilin-A coordinates priming and fusion of neurosecretory vesicles via intersectin

Sindhuja Gowrisankaran[1], Sébastien Houy [2,7], Johanna G. Peña del Castillo[1,7], Vicky Steubler[1,7], Monika Gelker [1], Jana Kroll [1], Paulo S. Pinheiro[2,3], Dirk Schwitters[1], Nils Halbsgut[1], Arndt Pechstein[4], Jan R.T. van Weering[5], Tanja Maritzen [4], Volker Haucke [4], Nuno Raimundo [6], Jakob B. Sørensen [2,8✉] & Ira Milosevic [1,8✉]

Endophilins-A are conserved endocytic adaptors with membrane curvature-sensing and -inducing properties. We show here that, independently of their role in endocytosis, endophilin-A1 and endophilin-A2 regulate exocytosis of neurosecretory vesicles. The number and distribution of neurosecretory vesicles were not changed in chromaffin cells lacking endophilin-A, yet fast capacitance and amperometry measurements revealed reduced exocytosis, smaller vesicle pools and altered fusion kinetics. The levels and distributions of the main exocytic and endocytic factors were unchanged, and slow compensatory endocytosis was not robustly affected. Endophilin-A's role in exocytosis is mediated through its SH3-domain, specifically via a direct interaction with intersectin-1, a coordinator of exocytic and endocytic traffic. Endophilin-A not able to bind intersectin-1, and intersectin-1 not able to bind endophilin-A, resulted in similar exocytic defects in chromaffin cells. Altogether, we report that two endocytic proteins, endophilin-A and intersectin-1, are enriched on neurosecretory vesicles and regulate exocytosis by coordinating neurosecretory vesicle priming and fusion.

[1] European Neuroscience Institute—A Joint Initiative of the University Medical Center Göttingen and the Max Planck Society Göttingen, Göttingen, Germany. [2] University of Copenhagen, Department for Neuroscience, Faculty of Health and Medical Sciences, Copenhagen, Denmark. [3] Center for Neuroscience and Cell Biology, University of Coimbra, Coimbra, Portugal. [4] Leibniz Research Institute for Molecular Pharmacology, Molecular Physiology and Cell Biology Section, Berlin, Germany. [5] Department of Clinical Genetics, Center for Neurogenomics and Cognitive Research, Amsterdam UMC, Amsterdam, The Netherlands. [6] Institute for Cellular Biochemistry, University Medical Center Göttingen (UMG), Göttingen, Germany. [7] These authors contributed equally: Sébastien Houy, Johanna G. Peña del Castillo, Vicky Steubler. [8] These authors jointly supervised this work: Jakob B. Sørensen, Ira Milosevic. ✉email: jakobbs@sund.ku.dk; imilose@gwdg.de

Release of vesicular content by exocytosis governs numerous biological events, including neurotransmission and neuro-modulation, which are mandatory for survival. Following exocytosis, endocytosis rapidly retrieves the exocytosed vesicle membrane and proteins. While the two processes are tightly coordinated, the molecular mechanisms underlying such coupling are not well understood[1–3].

Endophilin-A (henceforth endophilin), one of the best-characterized endocytic adaptors, is known to orchestrate various steps in clathrin-mediated endocytosis in mice, flies, and nematodes[4–12]. Endophilin acts as the hub of a protein network that coordinates membrane remodeling, cargo sorting, actin assembly, bud constriction as well as the recruitment of factors needed for fission and/or uncoating[13,14]. Endophilin was also proposed to have a central role in clathrin-independent endocytosis[15–17], and in ultrafast endocytosis[18]. Vertebrates express three endophilins encoded by three genes: SH3GL2 (endophilin 1, brain-enriched), SH3GL1 (endophilin 2, ubiquitous), and SH3GL3 (endophilin 3, brain- and testis-enriched). The three endophilin proteins have redundant functions and a similar structure: they contain an N-terminal Bin-Amphiphysin-Rvs (BAR)-domain that senses and induces membrane curvature, and a C-terminal SH3-domain that mediates protein interactions (e.g., with dynamins, synaptojanin-1)[4,5,12,19].

Along with the well-established role in endocytosis, several indications point to additional roles for endophilin at the neuronal synapse. (i) Endophilin was shown to undergo an association-dissociation cycle with the synaptic vesicles (SVs) and to be delivered to the periactive zone by exocytosis in the worm's nerve terminal[19]. (ii) Endophilin interacts with SV-resident vesicular glutamate transporter 1 (vGLUT1)[20,21], SV-associated synapsin[22], and intersectin-1[23], a conserved scaffold protein that was suggested to couple exocytosis and endocytosis[24,25]. In addition to its well-established role in clathrin-mediated endocytosis and actin dynamics, intersectin-1 has also been implicated in exocytosis, and it localizes at exocytic sites in neurosecretory PC12 cells and adrenal chromaffin cells[26–30]. (iii) The vGLUT1-endophilin-A1 interaction regulates the SV organization and spontaneous release at the murine hippocampal synapses[31]. (iv) Murine hippocampal neurons missing all three endophilins showed impaired SV recycling, a reduced number of SVs and altered neurotransmission[12]. However, it is not clear whether the impaired neurotransmission in neurons lacking endophilins results from defective endocytosis and SV recycling, or from endophilin's role in exocytosis, or both.

Here, we show that endophilin is directly involved in the regulation of exocytosis, independently of its endocytic roles. While exocytic and endocytic processes are tightly coupled at the neuronal synapse, such coupling is less prominent in neurosecretory cells, given that the large dense-core vesicles (LDCVs) originate from the trans-Golgi network and undergo a long maturation (minutes to hours) before they fuse with the plasma membrane[32]. Thus, we employed adrenal chromaffin cells, a well-established model to study calcium-regulated exocytosis, given that these cells use similar molecular machinery as neurons[33,34]. After LDCVs fuse with the plasma membrane in chromaffin cells, an orchestrated process of compensatory endocytosis efficiently removes the added membrane and proteins, and delivers them to a near-Golgi area, a process that takes tens of minutes[35]. We found that chromaffin cells, like neurons, contain mRNAs for all three endophilins. Thus, we employed chromaffin cells obtained from endophilin A1, A2, and A3 triple knock-out (TKO) mice described in Milosevic et al.[12] to decipher the role of endophilin in exocytosis.

## Results

### Endophilin-A is enriched at neurosecretory vesicles. Adrenal chromaffin cells contain numerous LDCVs that release their

content into the blood by fast exocytosis[34]. To check if endophilins 1–3 are expressed in chromaffin cells, we looked for the presence of the corresponding mRNAs and proteins. Firstly, RNA was isolated from cells extracted from the adrenal medulla obtained from wild-type (WT) and endophilin TKO P0 mice (TKO mice die a few hours after birth). All three endophilin mRNAs were detected by real-time quantitative PCR (qPCR) in the WT (endophilin 2 signal was the most prominent), but not in the TKO samples (Suppl. Fig. 1a). Next, western blots revealed the presence of endophilin 1 and endophilin 2 in the adrenal gland homogenate from WT mice, whereas these proteins were absent in the glands obtained from TKO mice (WT-wild-type, KOWTKO-endophilin $1^{-/-}$-$2^{+/+}$-$3^{-/-}$, TKO-endophilin $1^{-/-}$-$2^{-/-}$-$3^{-/-}$; Suppl. Fig. 1b). Endophilin 3, the least abundant member of the endophilin family in the brain[12], could not be detected by western blotting in the adrenal gland homogenate (Suppl. Fig. 1b).

Based on overexpression studies, endophilins are primarily cytosolic proteins that associate with membranes in various cells[12,36–38]. However, studies of native protein distributions are limited due to the lack of specific antibodies. We characterized two custom-made anti-endophilin antibodies and tested their specificity on the KO cells. These antibodies gave almost no staining in endophilin TKO chromaffin cells (Fig. 1a, b, right panels). Interestingly, in WT cells endophilin 1- and endophilin 2-specific signals were partially punctate and reminiscent of signals obtained with LDCV markers, e.g., chromogranin-A (CgA) (Fig. 1a, b, left panels). When chromaffin cells were co-immunostained for CgA and endophilin 1 (or endophilin 2, respectively), a significant colocalization between CgA and endophilins was detected (Suppl. Fig 1c, d; the values were corrected for accidental colocalization).

The enrichment of endophilin 1 and endophilin 2 on CgA-positive secretory vesicles was even more obvious on isolated plasma membranes generated by "unroofing" cultured chromaffin cells by a single sonication pulse[39,40]. Membrane sheets with attached secretory vesicles have been used previously to study the principles of vesicle release on the plasma membrane[41,42]. Plasma membrane sheets were subsequently co-immunostained for CgA and endophilins (Fig. 1c, d; endophilins are also involved in endocytosis at the plasma membrane, so it can be expected that a fraction of endophilin 1, or 2, do not colocalize with CgA-positive LDCVs). We further inspected the distribution of endophilin 1 and 2 in chromaffin cells upon stimulation (by 59 mM KCl buffer). Interestingly, a significant enrichment of endophilin 1 and endophilin 2-specific signals near/at the plasma membrane was detected upon stimulation (Fig. 1e, f, graphs below images show the line-intensity profiles; quantified in Fig. 1g, h).

In sum, endophilin-specific signal was present on majority, but not all CgA-labeled vesicles and it translocated to the plasma membrane upon stimulation. The occurrence of endophilin on LDCVs is unexpected, in particular in the light of two decades long research on this protein family. The function(s) of endophilin on neurosecretory vesicles and in exocytosis are entirely unknown.

### Endophilin-A promotes exocytosis in chromaffin cells. To investigate whether endophilins have a role in exocytosis, we performed fast measurements that combined membrane capacitance and amperometry recordings on chromaffin cells of endophilin TKO mice and corresponding littermate control (henceforth endophilin KOWTKO or KWK; note that a direct comparison to WT was not possible since the strain was constitutive knockout for endophilin 1 and 3, thus, C57BL6/J WT mice from a separate mouse line were used as an additional

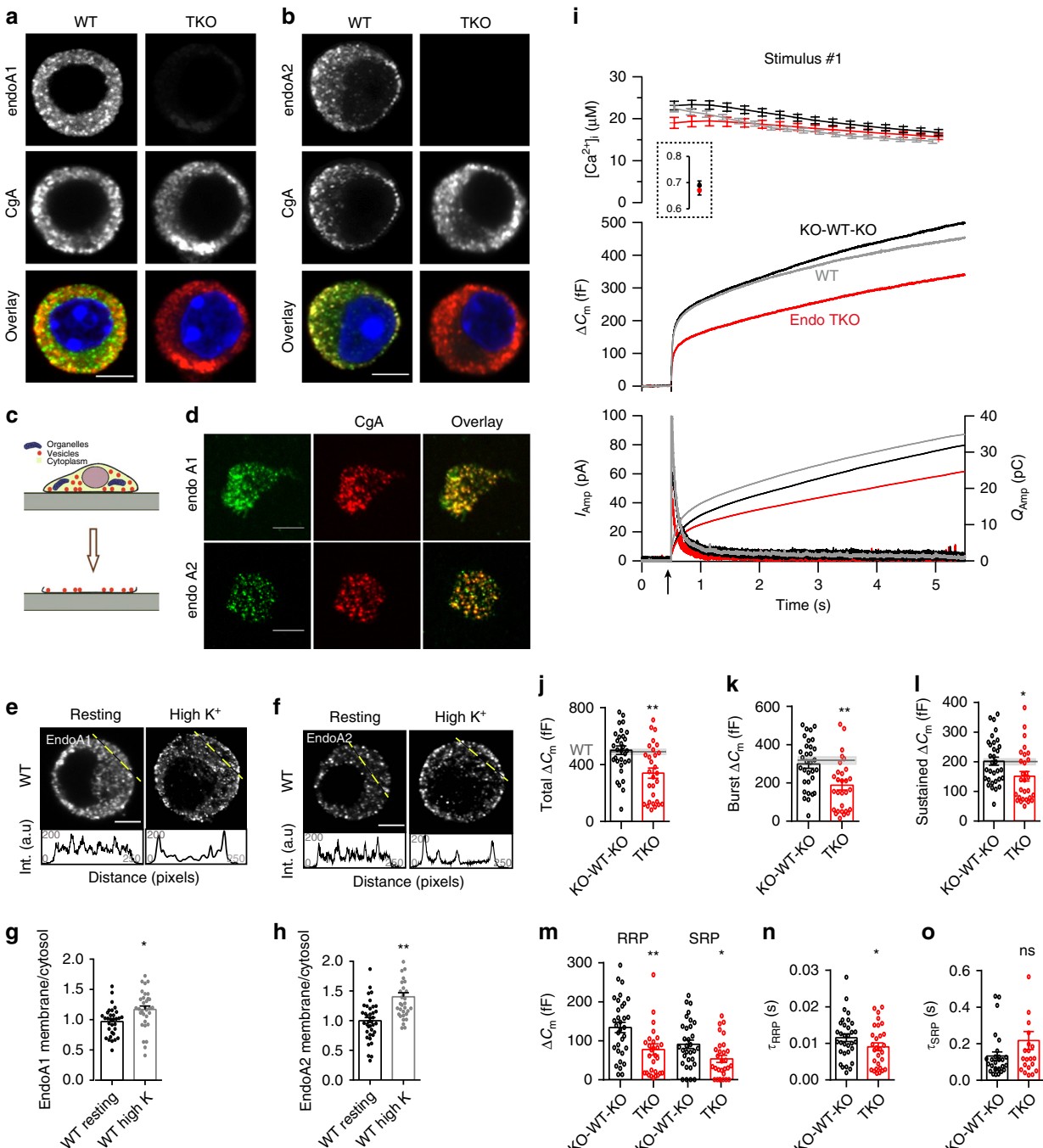

**Fig. 1 Endophilins are enriched at neurosecretory vesicles and required for efficient chromaffin cell exocytosis. a**, **b** Representative confocal images of WT and endophilin TKO mouse chromaffin cells stained for endophilin-1 (**a**) or endophilin-2 (**b**), and co-labeled with chromogranin-A (CgA), a LDCV marker. Scale bar 3 μm. For colocalization quantification, see Suppl. Fig 1c, d. **c** Plasma membrane sheet generation from cultured cells. **d** Representative plasma membrane sheets from chromaffin cells stained for endophilin-1 (top) or endophilin-2 (bottom) along with CgA. Scale bar 2 μm. Colocalization coefficient: 0.28 ± 0.13 endophilin-1 (N = 3 exp.; n = 26 sheets), 0.30 ± 0.11 endophilin-2 (N = 3 exp.; n = 20 sheets). **e**–**h** Upon stimulation, higher levels of endophilin-1 (**e**) and endophilin-2 (**f**) were present at the plasma membrane, as shown by intensity profiles along the lines indicated on the cells. Scale bar 3 μm. **g**, **h** Quantifications of endophilin redistribution. Endophilin-1: N = 3 exps; n = 32 (WT), 33 (TKO) cells, p = 0.005; Endophilin-2: N = 3 exps, n = 38 (WT) and 33 (TKO) cells, p < 0.0001, unpaired two-tailed t-test. (i) Exocytosis (indicated by the arrow) was reduced in endophilin TKO cells (red traces; N = 4 exps, 6 mice (29 cells)) compared to endophilin KOWTKO littermate cells (black traces; N = 4 exps, 6 mice (32 cells) and to (non-littermate) WT C57BL6/J cells (gray traces; N = 4 exps, 6 mice (60 cells)). **i** Top: intracellular calcium levels: The inset shows pre-stimulation calcium levels. Middle: averaged membrane capacitance changes upon Ca²⁺-induced exocytosis. Bottom: mean amperometric current (left axis) and cumulative charge (right axis). **j**–**l** Analysis of capacitance (KOWTKO: N = 4 exps, 6 mice (32) and TKO: N = 4 exps, 6 mice (29)) revealed an overall reduction of exocytosis. Gray line: the mean of 39 WT cells (non-littermate controls), shaded area: SEM. Note changes in burst (exocytosis within 1 s; **k**) and sustained phase of release (**l**). **m** Both RRP and SRP were reduced in TKO cells. **n** Fusion kinetics of RRP vesicles was faster in TKO cells. **o** Although on average slower, the SRP fusion kinetics constant was not significantly changed in TKO cells (**m**–**o**, N = 4, KOWTKO 6 mice (30) and TKO 6 mice (29)). Unpaired two-sided t-test, *p < 0.05, **p < 0.01, ns not significant. Error bars denote SEM.

control). The chromaffin cell cultures were prepared from adrenal glands at P0 and used between 2 and 4 days-in-vitro (DIV) for experiments. Capacitance and amperometry measurements were performed on a $Ca^{2+}$-calibrated setup as follows: each cell was loaded via a patch pipette with the photo-labile $Ca^{2+}$-chelator nitrophenyl-EGTA and the two $Ca^{2+}$-dyes (Fura-4F and Furapta) to enable accurate measurements of intracellular calcium concentrations ($[Ca^{2+}]_i$). Photolysis of caged $Ca^{2+}$ compound increased $[Ca^{2+}]_i$ from several hundred nanomolar to above 10 μM, resulting in robust exocytosis, which was assayed by an increase in membrane capacitance and amperometric current, one cell at a time. The measured increase in membrane capacitance is proportional to the change in chromaffin cell plasma membrane surface area, while simultaneous amperometric recordings allowed the measurement of secreted catecholamines during the exocytic events[34,39,43]. Remarkably, both capacitance and amperometry measurements from endophilin TKO chromaffin cells revealed reduced exocytosis ($340 \pm 36$ fF) when compared to the littermate control ($501 \pm 30$ fF for KOWTKO) and WT ($484 \pm 40$ fF) mice after the first stimulation (Fig. 1i, j, shaded gray area indicates WT). Notably, exocytosis from endophilin KOWTKO cells was comparable to that of WT cells originating from several different C57BL/6-based strains that have been recorded over the years[44–50].

Analysis of capacitance measurements revealed a reduction in the exocytic burst at 1 s following uncaging ($189 \pm 25$ fF in TKO vs. $299 \pm 22$ fF in KOWTKO and $315 \pm 16$ fF in WT, Fig. 1k) and the sustained component ($151 \pm 16$ fF in TKO vs. $202 \pm 14$ fF in KOWTKO and $176 \pm 12$ fF in WT; Fig. 1l). Further analysis of the exocytic burst revealed that both readily releasable pool (RRP) and slowly releasable pool (SRP) sizes (obtained from the amplitudes of triple-exponential fits to the capacitance traces) were significantly smaller ($78 \pm 15$ fF in TKO vs. $142 \pm 13$ fF in KOWTKO and $135 \pm 10$ fF in WT; $54 \pm 10$ fF in TKO vs. $91 \pm 11$ fF in KOWTKO and $97 \pm 7$ fF in WT, respectively; Fig. 1m). The fusion kinetics of the RRP vesicles from the TKO cells (time constants of the exponential fits) was sped up ($9.0 \pm 1.0$ ms in TKO vs. $12.1 \pm 1.0$ ms in KOWTKO and $10.9 \pm 0.9$ in WT; Fig. 1n). Curiously, we found that the RRP time constant correlated with the RRP size (Suppl. Fig. 1e; also noted in synaptotagmin-7 KO[47]). The kinetic parameter of the SRP was not significantly altered with the group size tested (Fig. 1o). Altogether, these data indicate that endophilin controls the size of the releasable pools and rate of RRP vesicle fusion. Similar results were found upon a second stimulation applied 100 s after the first stimulation (Suppl. Fig. 1f–i; a small exocytic response following the second stimulation in the TKO cells prevented reliable exponential fitting, thus the vesicle pool and kinetic analyses could not be performed). The ratio between second and first burst release was not changed (Suppl. Fig. 1j), suggesting that the vesicle pools, although smaller, could be efficiently refilled between the two stimuli.

To test if the exocytic phenotype is specific, and to examine the contribution of individual endophilins to chromaffin cell secretion, we re-introduced full-length endophilin 1, or endophilin 2 (Suppl. Fig. 2a), in endophilin TKO cells using a bicistronic lentiviral system, and performed electrophysiological measurements as before. First, we generated lentivirus and verified the expressions of endophilin 1 and 2 by western blotting (Suppl. Fig. 2b; HEK-293 cells were used here since lentivirus-infected primary chromaffin cells in culture do not yield enough material to perform a western blot). The lentivirus was then employed to infect primary chromaffin cells in culture: the co-expression of enhanced green fluorescent protein (EGFP) through the IRES system was used to identify transduced cells (Suppl. Fig. 2c; proteins were on average expressed at similar levels). Remarkably,

the expression of either endophilin 1 or endophilin 2 rescued the exocytic defects in TKO cells (Fig. 2a, b; note that the secretion from chromaffin cells derived from littermate controls was comparable to the secretion of C57BL6/J WT cells; Fig. 1i). Both burst and sustained exocytic components were efficiently rescued by either endophilins (burst: $221 \pm 28$ fF for endophilin 1 and $292 \pm 43$ fF for endophilin 2 vs. $137 \pm 15$ fF measured in endophilin TKO; sustained: $190 \pm 23$ fF for endophilin 1 and $178 \pm 20$ fF for endophilin 2 vs. $92 \pm 24$ fF in endophilin TKO; Fig. 2c, d; shaded gray area indicates littermate control), and so were the RRP size (Fig. 2e) and the fusion kinetics of the RRP (Fig. 2f). The kinetics of the SRP were unchanged when either endophilin 1 or 2 were expressed (Fig. 2g). Similar results were noted upon a second stimulation (Suppl. Fig. 2d–g). Notably, the releasable pools were recovered efficiently between two stimuli that were 100 s apart, as revealed by an unchanged ratio between second and first burst (Suppl. Fig. 2h). Altogether, these data reveal that the effect of endophilin on exocytosis was specific, and that the expression of either endophilin 1 or 2 was sufficient to support exocytosis in chromaffin cells.

To inspect whether chromaffin cells without endophilin show changes in single-vesicle fusion events, we performed a thorough examination of single amperometric spike parameters from endophilin TKO cells and compared them to TKO cells expressing endophilin 2. Here, secretion was elicited by loading chromaffin cells in whole-cell mode with low-calcium intra-pipette solution, as detailed in Pinheiro et al.[48]. Representative amperometric traces for endophilin TKO and endophilin TKO expressing endophilin 2 are shown in Fig. 2h; the analyzed properties of single spikes are illustrated in Fig. 2i. Significant differences were observed in the number of detected events per cell (Fig. 2j), single spike charge (calculated as the time integral of the amperometric current; reflects the total amount of catecholamines oxidized at the electrode) and amplitude (Fig. 2k, l)—all these parameters were found to be decreased in endophilin TKO cells, while the kinetic features of single spikes were not changed (Fig. 2m–o). Amperometric spikes are often preceded by a pre-spike foot that reflects catecholamine leakage through the forming fusion pore[51]. The pre-spike foot duration was not altered, but its amplitude was smaller in endophilin TKO cells (Fig. 2p, q), further supporting the notion that endophilin participates in LDCV fusion.

In addition to the capacitance measurements that allow analysis of exocytosis of the whole chromaffin cell on the millisecond time scale and the amperometric measurements that report directly on catecholamine release, we employed an imaging-based assay that capitalizes on synaptotagmin-7-pHluorin (Syt-7-pHluorin) and synaptotagmin-1-pHluorin (Syt-1-pHluorin; pH-sensitive LDCV-resident probes) to independently inspect exocytosis in this model system (Suppl. Fig. 2i). We expressed Syt-7-pHluorin, or Syt-1-pHluorin, in chromaffin cells using lentivirus and assayed vesicle release upon stimulation (59 mM KCl), and in the presence of the endocytosis inhibitor Pitstop-2 (Suppl. Fig. 2j–n)[52]. We found that endophilin TKO cells expressing Syt-7-pHluorin (Suppl. Fig. 2j–k), or Syt-1-pHluorin respectively (Suppl. Fig. 2l–n), showed a lower number of fluorescent puncta that appeared upon stimulation compared to WT and littermate controls, thereby confirming that the loss of endophilins in chromaffin cells impairs exocytosis.

Taken together, these data strongly suggest that endophilin has a direct role in exocytosis by regulating the fusion of secretory vesicles and the size of the releasable vesicle pool.

**LDCV numbers are not altered in the absence of endophilin-A.** The decreased exocytosis in endophilin TKO chromaffin cells

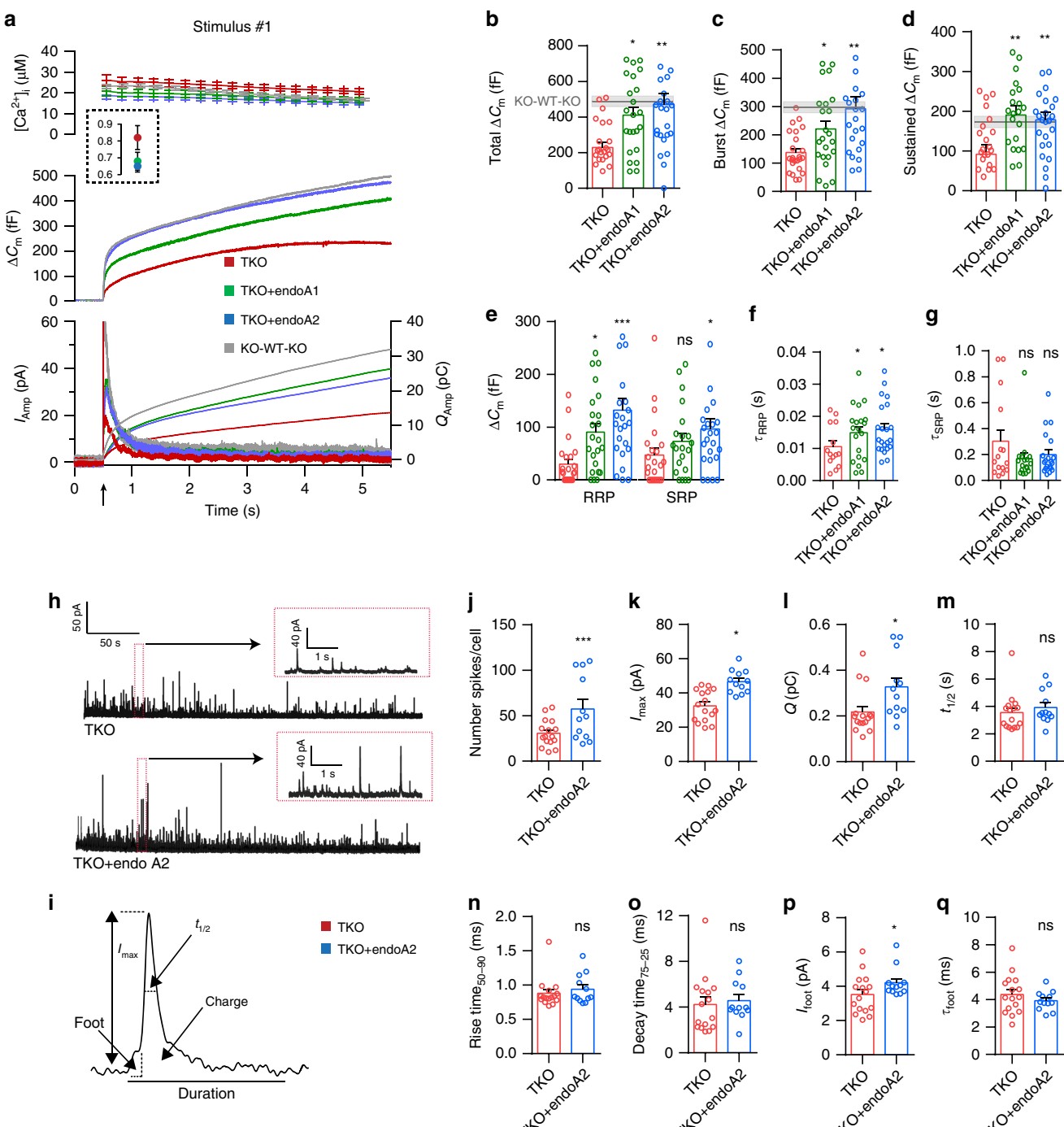

**Fig. 2 Expression of endophilin 1 and endophilin 2 rescued exocytosis in endophilin TKO cells. a**, **b** Expression of endophilin 1 and endophilin 2 full-length rescued the exocytic defects seen in endophilin TKO cells. Panel arranged as in Fig. 1i, with three groups: endophilin TKO (red traces; $N = 4$, 4 mice (24 cells)), TKO + endophilin 1 (green traces; $N = 4$, 4 mice (23 cells)) and TKO + endophilin 2 (blue traces; $N = 4$, 4 mice (25 cells)). Control KOWTKO data from Fig. 1I (gray trace; $N = 4$, 6 mice (32 cells)) are superimposed. Note that both endophilin 1 and endophilin 2 can rescue exocytosis; rescue with endophilin 2 is indistinguishable from control KOWTKO cells. **c**–**e** Burst and sustained component, as well as RRP size, were rescued upon expression of endophilin 1 and 2, respectively. **f**, **g** The altered kinetics of the RRP in TKO was rescued (**f**), while the time constant of the SRP was not significantly changed (**g**) upon expression of endophilin 1 and 2, respectively (**b**–**g**) Kruskal–Wallis test with Dunn's multiple comparison test. **h** Exemplary traces from amperometric recordings of endophilin TKO and endophilin TKO expressing endophilin 2. Insets show magnified view. **i** Schematic of analyzed amperometric spike parameters. **j**–**q** Amperometry analysis (210 s recording per cell) reveals problems in vesicle fusion: number of fusion events per cell (**j**), single spike amplitude (**k**) and charge (**l**) were significantly decreased in endophilin TKO cells, while the kinetics of single fusion events, such as duration at half-maximal amplitude (**m**), rise time (**n**), and decay time (**o**), was unchanged. The stability of the fusion pore was also altered, as shown by shorter foot amplitude (**p**) while the foot duration (**q**) was not changed. TKO: 2 mice (16 cells), TKO + endophilin: 2 mice (12 cells)—each cell is a biological replicate. Error bars denote SEM. $N =$ number of independent replicates. Unpaired two-sided $t$-test. $^*p < 0.05$, $^{**}p < 0.01$, $^{***}p < 0.001$, ns not significant.

suggest that these cells might have less LDCVs able to undergo exocytosis, or that the exocytic process itself is affected, or both. To distinguish between these possibilities, we performed comprehensive ultrastructural studies by electron microscopy (EM) and 3D-confocal imaging.

Ultrastructural EM analyses were performed on cultured chromaffin cells and adrenal glands. Independently of whether chromaffin cells were analyzed in culture or gland, we noted that the overall LDCV morphology in endophilin TKO cells was comparable to controls (WT and littermate endophilin KWK cells) (Fig. 3a and Suppl. Fig. 3a). The average number of LDCVs per cell, as well as their sizes, were not significantly different between endophilin TKO, control littermates and WTs (Fig. 3b, c and Suppl. Fig. 3b–d). This observation rules out the possibility that endophilin affects exocytosis by altering the number of LDCVs. The number of docked vesicles was also not altered (Fig. 3d), nor was the distribution (distance from the plasma membrane) of LDCVs in endophilin TKO cultured cells in comparison to controls (Fig. 3a bottom panels, Fig. 3e, f). In chromaffin cells from adrenal glands, however, LDCVs were distributed further away from the plasma membrane in comparison to controls (Suppl. Fig. 3e–g), and less LDCVs were found within 10 nm from the plasma membrane (Suppl. Fig. 3f). Yet, this mild phenotype was observed only in chromaffin cells in intact glands, and not in the cultured chromaffin cells, where the exocytic defect was seen by electrophysiological measurements.

The ultrastructural analysis was further complemented by two independent approaches: 3D-confocal microscopy and western blotting. First, WT, endophilin KOWTKO, and TKO chromaffin cells were immunostained with the cargo marker CgA, and the whole cell (acquired through z-stacks) was imaged by the Zeiss Airyscan confocal system. Representative images are shown in Fig. 4a. Quantification of CgA-positive LDCVs in the whole-cell volume revealed similar numbers of LDCVs in endophilin TKO and controls (Fig. 4b). Concordantly, the levels of CgA protein were not altered in endophilin TKO adrenal gland homogenates (Suppl. Fig. 4a, b).

Altogether, the reduced exocytosis in chromaffin cells without endophilin was not a result of the altered vesicle number, morphology or cargo (i.e., CgA) amount.

**The exocytic protein machinery is not altered in chromaffin cells without endophilin-A.** The reduced exocytosis in endophilin TKO cells occurs at the milliseconds-to-seconds time scale. Endophilins, being major endocytic proteins, could have an indirect effect on the LDCV composition and/or membrane and protein recycling processes in chromaffin cells. Despite the overall abundance of LDCVs and the unaltered morphology of individual LDCVs in endophilin TKO chromaffin cells, it is possible that some LDCVs may not be able to undergo exocytosis due to changes in accessory factors required for exocytosis. Specifically, SNAREs, Sec1/Munc18 (SM)-proteins, Munc13s and other exocytic proteins may not be efficiently recycled from the previous rounds of exocytosis, or properly displayed on LDCVs in the absence of endophilin. Therefore, we checked the abundance and distribution of key exocytic proteins in endophilin TKO cells by immunocytochemistry (ICC), western blotting, and real-time qPCR.

As an exemplary exocytic protein, we first analyzed the abundance and distribution of synaptotagmin-1 (Syt-1), a $Ca^{2+}$-sensor important for LDCV exocytosis in chromaffin cells (Fig. 4c–e). We immunostained endophilin TKO, littermate control KOWTKO and WT cells for Syt-1, and noticed that neither distribution (Fig. 4c, d), nor protein levels (Suppl.

Fig 4a, b), of Syt-1 were altered in endophilin TKO cells. We next checked whether proper amounts of Syt-1 are present on LDCVs by quantifying the intensity of Syt-1 signal on the CgA-positive structures (through ICC). The analysis showed no statistical difference of Syt-1 intensity on CgA-positive puncta between WT and endophilin TKO cells (Fig. 4e).

We further inspected protein levels of Syt-1, synaptophysin, key SNAREs (SNAP-25, syntaxin-1, synaptobrevin-2/VAMP2) and Munc18-1 in adrenal gland homogenates by western blotting, and found no significant difference between endophilin TKO cells and controls (littermate endophilin KOWTKO and WT samples were used as controls; Suppl. Fig. 4a–d). Also, a number of exocytic proteins, namely SNAP-25, synaptobrevin-2/VAMP2, Munc18-1 and Syt-7 revealed no significant changes by immunofluorescence in protein level and distribution in endophilin TKO chromaffin cells (Fig. 4f, g).

We also inspected the mRNA levels of several genes encoding cargo, exocytic proteins and proteins known to play a role in LDCV formation and/or maturation. RNA was isolated from adrenal medulla obtained from WT and endophilin TKO P0 mice, and subjected to real-time qPCR. No difference was observed in the mRNA levels of chromogranin-B and dopamine-ß-hydroxylase (DBH; LDCV cargo proteins), Syt-1, synaptotagmin-4, synaptobrevin-2/VAMP2 and VAMP4 (LDCV-resident proteins), vacuolar ATPase subunit $V_0a$, syntaxin 6 and syntaxin 16 (important for LDCV formation and/or maturation) (Suppl. Fig. 4e).

Taken together with the unchanged number of LDCVs, these data suggest that the reduced catecholamine release in the absence of endophilin is likely a consequence of endophilin's direct action in exocytosis.

**Endocytic defects in chromaffin cells without endophilin-A.** Endophilin TKO cells show an unaltered number of LDCVs, a normal distribution and abundance of the key exocytic proteins, yet, it is possible that altered endocytosis may affect exocytosis in chromaffin cells. While exocytosis is well studied in this model system, endocytic modes are far from understood. Two temporally and mechanistically distinct forms of endocytosis have been reported: rapid endocytosis that depends on dynamin-1 and GTP, and slow endocytosis that involves dynamin-2 and clathrin[53,54]. We predominantly studied slow (presumably clathrin-mediated) endocytosis since it is not known to which extent these cells undergo fast local recycling.

We examined the protein levels and distributions of the main endocytic factors, namely clathrin heavy chain (HC), adaptor protein 2 (AP2), adaptor protein 180 (AP180), dynamins 1, 2, and 3 (detected by pan-anti-dynamin antibody) by immunofluorescence analysis and western blotting. Except for clathrin-HC whose levels were mildly elevated in the absence of endophilin (by ICC analysis only, Fig. 5a), we detected no difference in the overall levels of the aforementioned proteins, both by quantifying immunofluorescence in chromaffin cells (Fig. 5a) and western blotting of adrenal gland homogenates (Fig. 5b). In addition, the distribution of AP2 and dynamin was unaltered in endophilin TKO cells (Fig. 5a).

The slow endocytic recycling process in chromaffin cells was tested by three approaches: the uptake of transferrin-Alexa Fluor[TM] 546 (A546; clathrin-dependent), membrane marker mCLING-Atto647 and recombinant cholera toxin subunit B (CT-B)-Alexa Fluor 594 (A594; clathrin-dependent and -independent). First, we looked at the 10 min-uptake of transferrin-A546 by analyzing whole-cell volume of endophilin TKO, littermate control KOWTKO and WT cells. There was no significant difference between WT and endophilin TKO cells, while, uncharacteristically, endophilin KOWTKO cells displayed

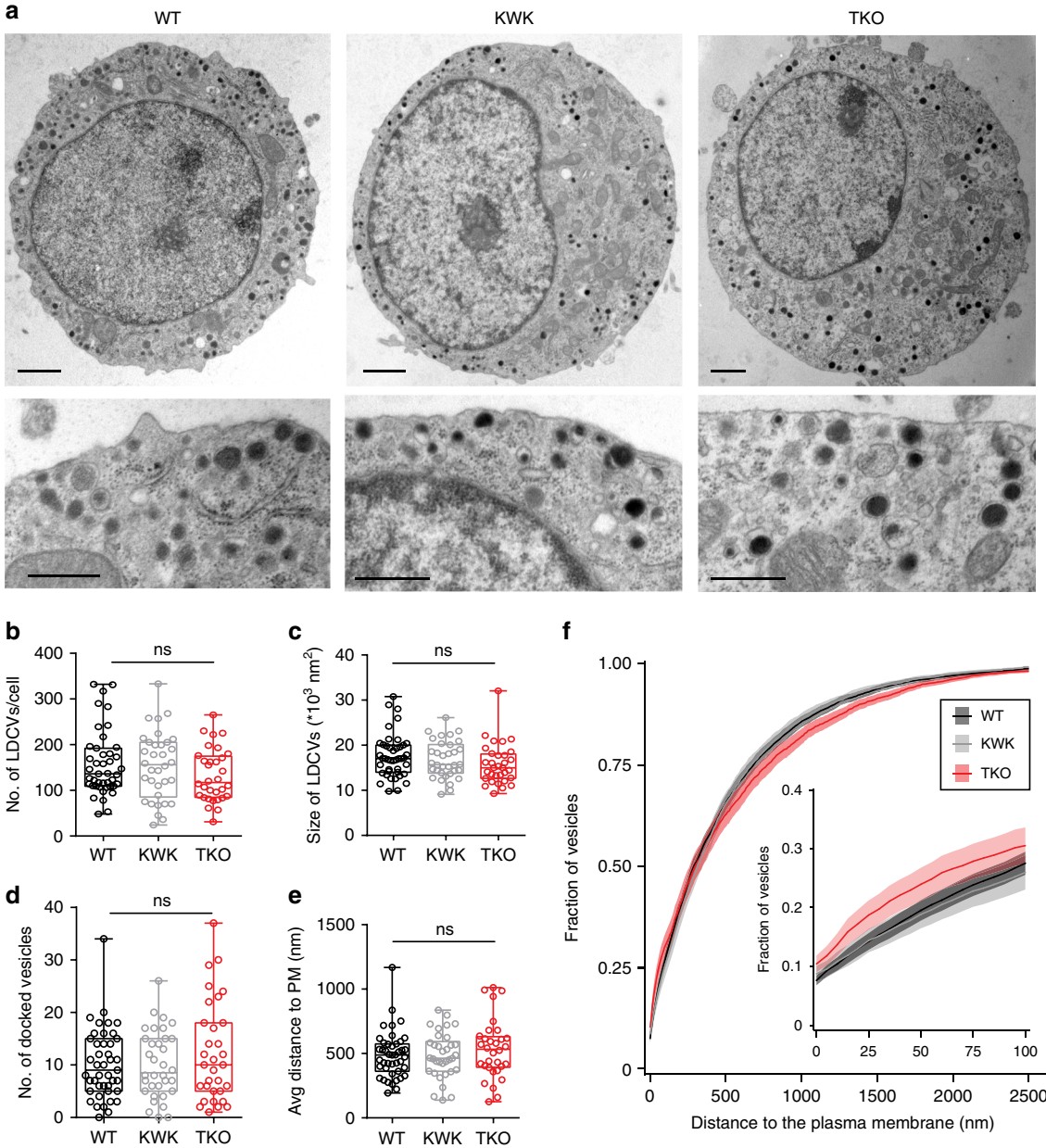

**Fig. 3 Number, distribution, and size of LDCVs were not altered in endophilin TKO chromaffin cells in culture. a** Typical electron micrographs of WT, endophilin KWK (littermate control) and endophilin TKO chromaffin cells in dissociated culture. Bottom row shows the sub-plasma membrane area at higher magnification. Scale bar: 1 µm (top row) and 0.5 µm (bottom row). **b** Total number of vesicles/cell cross section, **c** surface of the vesicle projection, **d** number of docked vesicles/cell cross section and **e** the average distance of vesicles to the plasma membrane in dissociated chromaffin cell cultures. Bar shows mean, error bars SEM, dots are individual measurements. **f** Relative cumulative distribution of vesicles to the plasma membrane in 5 nm bins. Insert shows the distribution profile in the first 100 nm, line shows average, ribbon shows SEM (Kolmogorov–Smirnov test with Bonferroni-correction). **a–f** Three independent experiments, four mice each, cells WT (43), KOWTKO (34), and TKO (32). One-way ANOVA after Tukey's post-hoc test, ns not significant.

higher levels of internalized transferrin-A546 than WT and TKO (Fig. 5c, d). We further inspected the uptake of mCLING-Atto647 that binds to the plasma membrane and whose internalization can be readily monitored for minutes[55] (Suppl. Fig. 5a and Supplementary Movie 1 shows stimulated chromaffin cells; for more information on the assay characterization see Supplemental data). We first characterized the specificity of mCLING uptake in chromaffin cells by stimulating cells with high potassium in the presence of Pitstop-2 (clathrin coat formation inhibitor) and found that this inhibitor blocked the uptake of mCLING efficiently (Suppl. Fig. 5b and Supplementary Movie 2—note that the cell surface increases upon stimulation since endocytosis was

blocked). When mCLING-Atto647 was applied to chromaffin cells without endophilin, a mild uptake delay was observed in the first 2–3 min, but at 8 min similar numbers of internalized vesicles (mCLING-positive structures) were detected in endophilin TKO and controls (WT and endophilin KOWTKO cells) (Fig. 5e, f; Supplementary Movie 3 and Supplementary Movie 4). A comparable result was obtained with the uptake of CT-B-A594, which is also internalized by both clathrin-mediated and clathrin-independent endocytosis (Suppl. Fig. 5c, d). In summary, these data indicate that potential endocytic defects in the absence of endophilin cannot account for the observed effect on exocytosis in this model system.

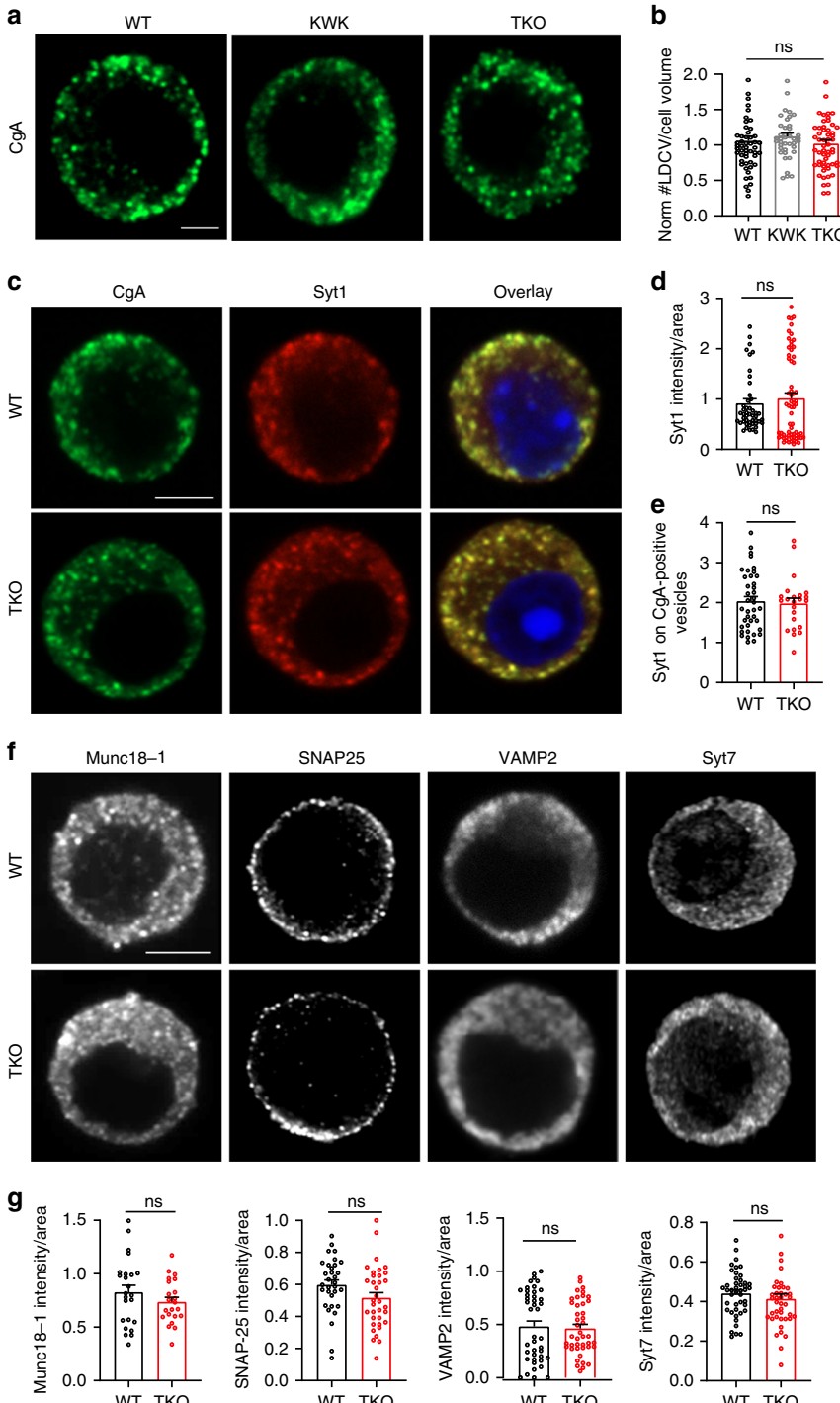

**Fig. 4 Exocytic machinery is unaltered in the chromaffin cells without endophilin. a, b** Confocal images of endophilin TKO and control (WT and littermate endophilin KOWTKO) chromaffin cells stained for chromogranin-A (CgA; LDCV marker) revealed no difference in the number of CgA-positive vesicles measured in the whole volume of the cell ($N = 4$, WT 5 mice (54 cells), KOWTKO 4 mice (42 cells), and TKO 5 mice (57 cells), $p$-value $= 0.96$, one-way ANOVA with Tukey's multiple comparison). An optical section through the equatorial plane of the cells is shown in a. Scale bar 3 μm. **c** Images of endophilin TKO, littermate KOWTKO and WT chromaffin cells stained with anti-synaptotagmin-1 (Syt-1) and CgA antibodies. Scale bar 3 μm. **d** Immunofluorescence levels revealed no change in synaptotagmin-1 levels ($N = 4$, WT 5 mice (48 cells) and TKO 5 mice (67 cells), $p$-value $= 0.45$, unpaired two-sided $t$-test). **e** Quantification of the Syt-1 intensity on the CgA-positive vesicles revealed no significant difference between the genotypes ($N = 4$, WT 4 mice (39 cells) and TKO 3 mice (23 cells), $p$-value $= 0.76$, unpaired two-sided $t$-test). **f** Representative images of endophilin TKO and WT chromaffin cell stained for Munc18-1, SNARE proteins SNAP25 and VAMP2 and synaptotagmin-7. **g** Quantification of Munc18-1 ($N = 3$, WT 3 mice (24 cells) and TKO 3 mice (22 cells), $p$-value $= 0.25$, SNAP-25 ($N = 3$, WT 3 mice (32 cells) and TKO 3 mice (35 cells), $p$-value $= 0.07$), VAMP2 ($N = 3$, WT 4 mice (43 cells) and TKO 4 mice (43 cells), $p$-value $= 0.27$) and Syt7 ($N = 3$, WT 3 mice (44 cells) and TKO 3 mice (42 cells), $p$-value $= 0.082$) immunofluorescence showed no significant changes for any of these proteins. Error bars denote SEM. $N =$ number of independent replicates. ns not significant.

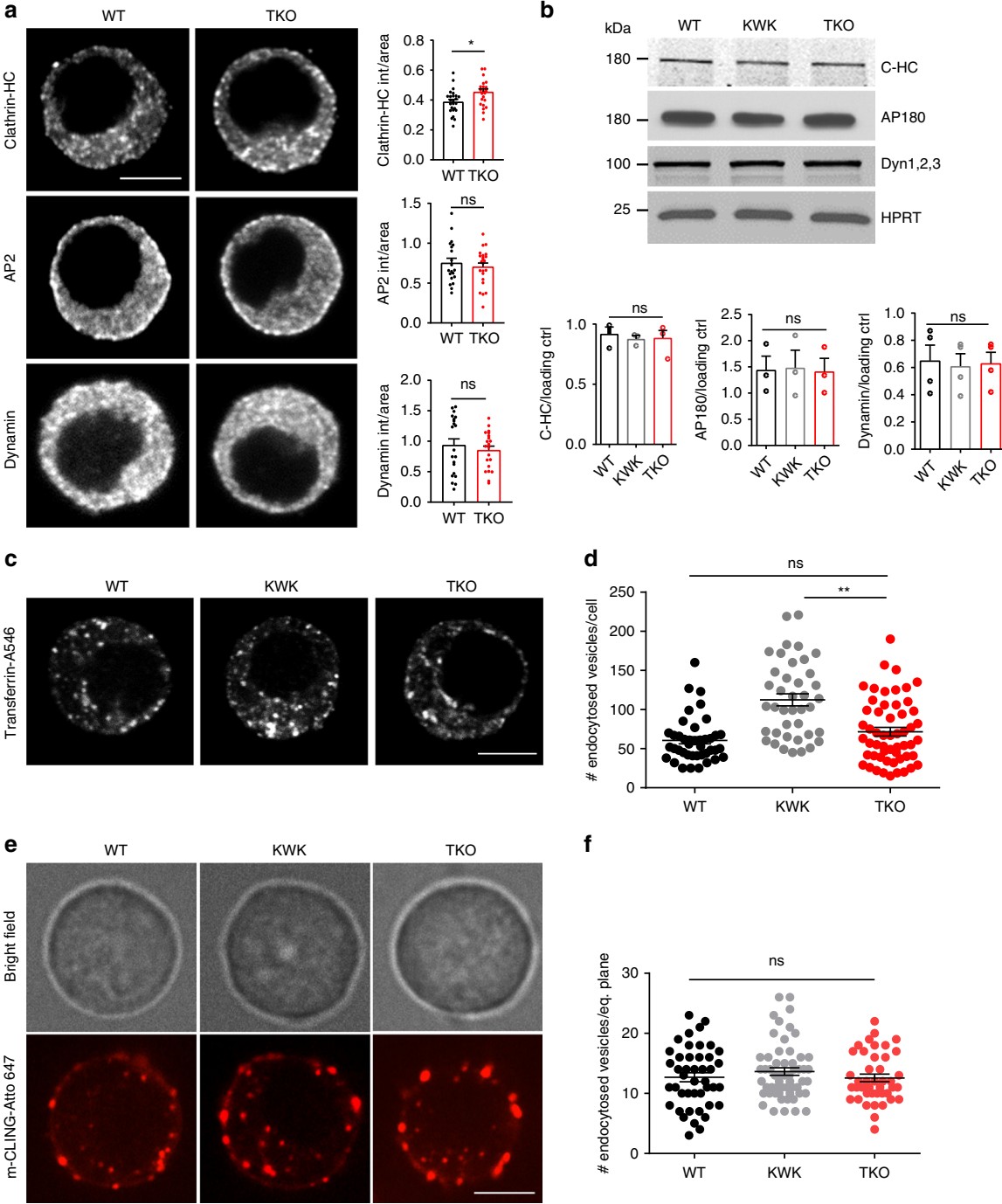

**Fig. 5 Minor endocytic defects in the absence of endophilin in chromaffin cells. a** Immunofluorescence for clathrin heavy chain (HC), adaptor protein 2 (AP2) and dynamin-1 in WT and endophilin TKO cells. Scale bar 3 μm. Fluorescence quantification revealed small but significant increase in clathrin-HC intensity ($N = 3$, WT 3 mice (25 cells) and TKO 3 mice (24 cells)), whereas intensities of AP2 ($N = 3$, WT 3 mice (20 cells) and TKO 3 mice (23 cells)) and dynamin-1 (N = 3, WT 3 mice (21 cells) and TKO 3 mice (21 cells)) were unaltered in endophilin TKO cells. **b** Protein levels of the main endocytic factors—clathrin, adaptor protein 180 (AP180) and dynamins 1–3 (inspected by western blotting) were not altered in adrenal gland homogenates. Below: Quantification from clathrin-HC ($N = 3$, 3 samples/genotype), AP180 ($N = 3$, 3 samples/genotype) and dynamin ($N = 4$, 4 samples/genotype)—note that each sample on the blot originates from 6 to 8 glands from 3 to 4 mice of the same genotype pooled together. **c, d** Transferrin (conjugated with Alexa Fluor-546) uptake in the endophilin TKO cells compared to the littermate control and WT ($N = 3$, 3 mice per genotype WT (41 cells), KOWTKO (40 cells), and TKO (55 cells); note that uptake was analyzed in the whole cell, no difference in uptake between WT and endophilin TKO cells was observed). Scale bar 3 μm. **e, f** mCLING-Atto647 uptake (example cells shown in Supplementary Movies 1, 3, and 4) by chromaffin cells of indicated genotypes showed no significant difference in the number of endocytosed vesicles ($p$-value = 0.44, $N = 3$, 3 mice per genotype: WT (44 cells), KOWTKO (56 cells), and TKO (42 cells)). Note that uptake was analyzed in only one cell plane, and the bright-field image was acquired before recording the mCLING fluorescent signals so the focal planes are not always identical. The specificity of mCLING-Atto647 uptake was tested in the stimulated cells in the presence of Pitstop-2 inhibitor (Supplementary Movie 2 and Suppl. Fig. 5b). Error bars denote SEM. N number of independent replicates. One-way ANOVA after Tukey's post-hoc test, *$p < 0.05$, **$p < 0.01$, ns not significant.

**Endophilin's BAR-domain is not sufficient to mediate exocytic release from chromaffin cells.** To get mechanistic insights on how endophilin regulates the exocytic process, we looked at the function of endophilin domains: BAR-domain and SH3-domain. In nematodes, it has been shown that endophilin's BAR-domain is necessary and sufficient to mediate its role in endocytosis[19], while in mammalian cells both domains were needed[12]. We first tested whether endophilin's BAR-domain alone (i.e., endophilin without SH3-domain) is sufficient to support exocytosis in chromaffin cells. We expressed endophilin 1 BAR-domain and endophilin 2-BAR-domain respectively (Fig. 6a; expressed with EGFP through the bicistronic IRES-expression system: the expression levels of all tested proteins were comparable; Suppl. Fig. 6e) in endophilin TKO cells, and performed capacitance and amperometry measurements as before. Interestingly, expression of endophilin 1 or endophilin 2-BAR-domain did not result in a rescue but rather in a further reduction of secretion from endophilin TKO cells, either during the first (Fig. 6b–e) or the second stimulus (Suppl. Fig. 6a–d). The small responses revealed an overall decrease in exocytosis, including both burst and sustained component (Fig. 6c–e). This dominant-negative effect reveals that the SH3-domain-mediated functions are important for endophilin's role in exocytosis, and that the full-length protein is needed to support exocytosis in chromaffin cells.

**Endophilin's role in exocytosis is mediated via intersectin-1.** Endophilin's SH3-domain is known to mediate its interaction with several proteins, yet only two of them have been implicated in chromaffin cell exocytosis: dynamins[56–58] and intersectin-1[26–28,30]. Distribution and levels of dynamins were not altered (Fig. 5a, b), as detailed before. Curiously, while levels of intersectin-1 (ITSN-1), a membrane-associated protein that coordinates exocytic and endocytic vesicle traffic, were comparable (Suppl. Fig 7a, b; both short and long isoform of ITSN-1 are shown), the distribution of ITSN-1 was altered in chromaffin cells lacking endophilins (Fig. 7a; the line-intensity profile through the depicted cells is shown below the images). Upon stimulation, ITSN-1 was recruited to the plasma membrane in WT cells (Supplementary Movie 5 and Fig. 7a), as originally reported by Malacombe et al.[26]. However, this redistribution did not happen in endophilin TKO cells (Fig. 7a, b). A detailed examination in resting chromaffin cells revealed that the fraction of ITSN-1 on the plasma membrane was higher in the TKO compared to the WT (Fig. 7b and Suppl. Fig. 7c, d). Similar observations were made with intersectin-2 (Fig. 7c, d and Suppl. Fig. 7e, f). In sum, the cellular distribution of intersectin-1 and intersectin-2 in chromaffin cells was altered (1) after stimulation and (2) in the absence of endophilins.

To check if this effect is specific, we attempted to rescue the ITSN-1 distribution by expressing either endophilin 1 or endophilin 1-ΔITSN (endophilin 1 E329K + S336K mutant that does not bind ITSN-1[23]) in endophilin TKO cells. Upon endophilin 1 expression in TKO cells, the distribution of ITSN-1 resembled the protein distribution in WT cells (as detected by immunostaining—Fig. 7e, compare to Fig. 7a, quantification in Fig. 7f). Curiously, the expression of mutant endophilin 1-ΔITSN did not have the same effect, and ITSN-1 was still mislocalized (Fig. 7e, f). These data suggest that the endophilin-intersectin interaction is important for intersectin's distribution in chromaffin cells and that it regulates intersectin's access to the plasma membrane where vesicle priming and fusion happens.

Given that the expression of endophilin 1 in endophilin TKO cells rescued exocytosis when inspected by combined fast capacitance and amperometry measurements (Fig. 2a–g), we tested whether the same effect could be achieved by expressing the

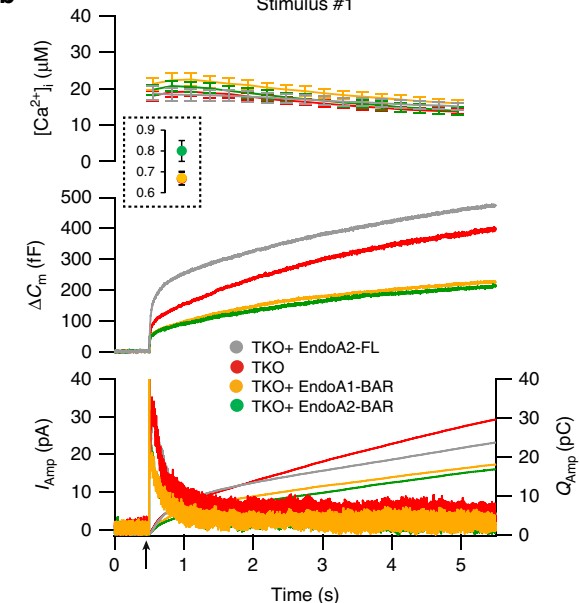

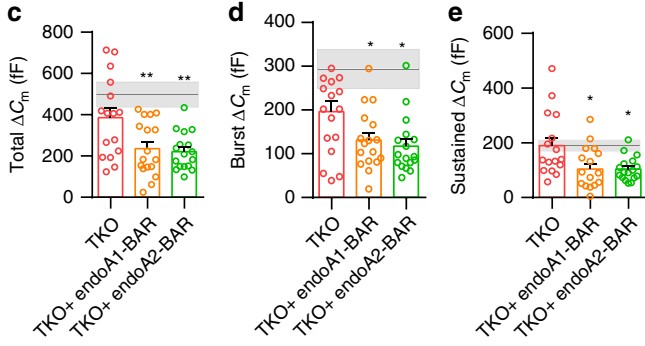

**Fig. 6 Endophilin BAR-domains are not sufficient to mediate chromaffin cell exocytosis. a** Schematic depicting the domain structure of endophilin 2 and two mutants lacking the SH3 domain expressed in the subsequent experiments. **b** Exocytosis induced by calcium uncaging in endophilin TKO chromaffin cells (red traces) compared to TKO cells expressing either endophilin 1 BAR domain (yellow traces), endophilin 2-BAR domain (green traces) or full-length endophilin 2 WT protein (gray traces). Panel arranged as in Fig. 1i, top: intracellular calcium level increase induced by flash photolysis at 0.5 s (at arrow). The inset shows the pre-flash calcium levels. Middle: averaged traces of membrane capacitance upon Ca²⁺-induced exocytosis. Bottom: mean amperometric current (left axis) and cumulative charge (right axis). **c–e** Quantification of changes in capacitance revealed a further reduction in different phases of release (burst and sustained) in TKO cells expressing endophilin 1-BAR or endophilin 2-BAR domain. Gray line indicates the mean of TKO cells expressing endophilin 2 full-length protein and the shaded area indicates the SEM. **b–e** $N = 3$, 3 mice/condition, cells TKO (16), TKO + endophilin1 BAR (17) and TKO + endophilin 2-BAR (17). Error bars denote SEM. $N =$ number of independent replicates. One-way ANOVA with Tukey's multiple comparison test, *$p < 0.05$, **$p < 0.01$.

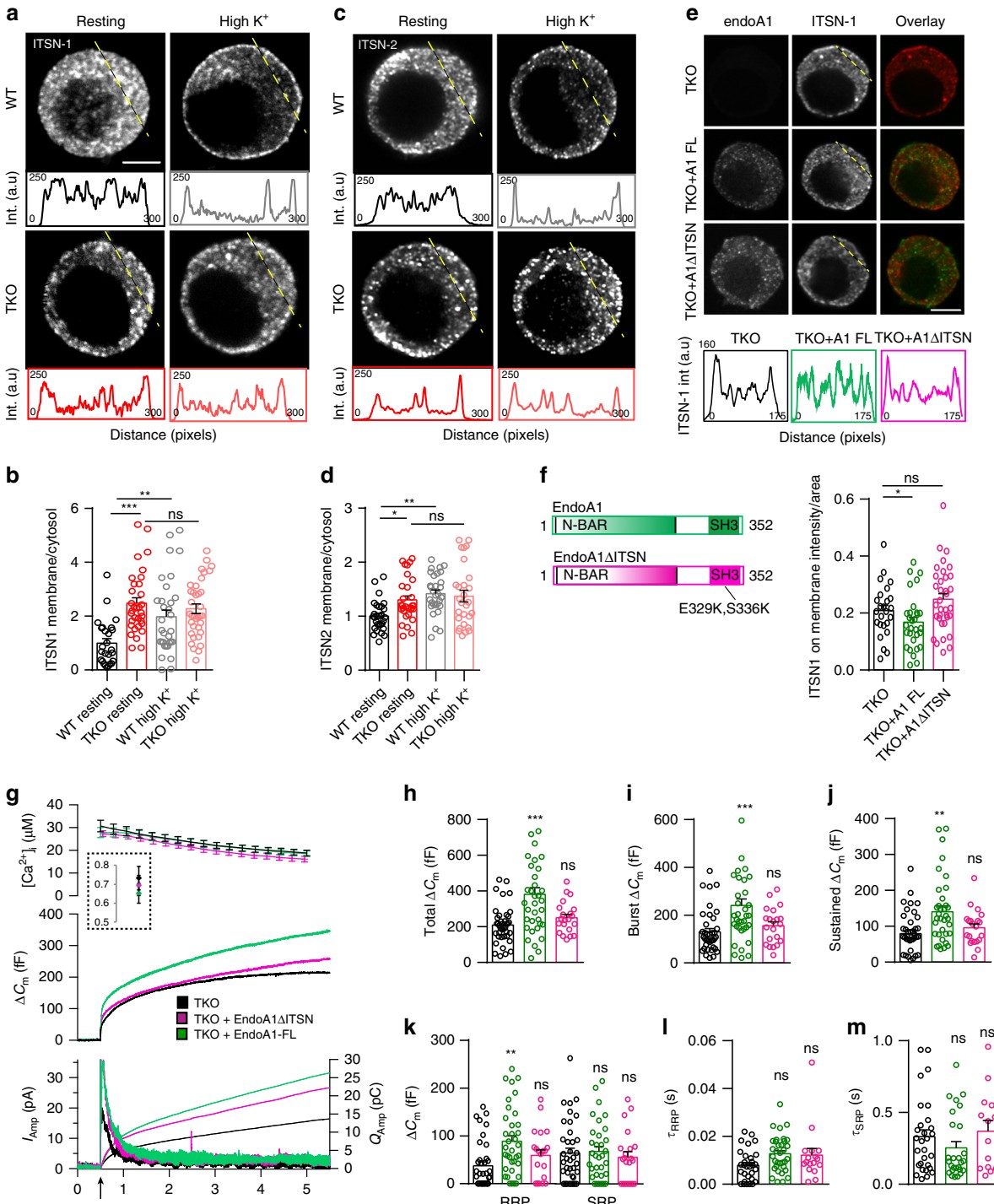

**Fig. 7 Endophilin's role in exocytosis is mediated, at least in part, through its interaction with intersectin-1. a–d** Distribution of ITSN-1 and ITSN-2 was altered in the endophilin TKO cells. Representative confocal images of chromaffin cells stained for ITSN-1 (**a**) and ITSN-2 (**c**) under resting and stimulated (depolarization by high K⁺) conditions (note that different cells are shown for each condition in the panel). Intensity line profiles below indicate ITSN-1 and ITSN-2 intensities along the line (the approx. line position is marked in yellow in **a** and **c**). Quantification of ITSN-1 (**b**) and ITSN-2 (**d**) intensities in the cytosol vs. near the membrane (see Methods and Suppl. Fig. 7) revealed an altered distribution of ITSN-1 and ITSN-2 in endophilin TKO cells that did not further change upon stimulation. ITSN-1 ($N = 3$, 3 mice each, WT resting (26 cells), TKO resting (33 cells), WT high K (32 cells) and TKO high K (36 cells), and ITSN2 ($N = 3$, 3 mice/condition, WT resting (29 cells), TKO resting (30 cells), WT high K (30 cells) and TKO high K (28 cells). **b**, **d** note that WT resting vs. WT high K⁺, WT resting vs. TKO resting and TKO resting vs. TKO high K⁺ were compared). **e**, **f** The altered distribution of ITSN-1 in endophilin TKO cells was rescued by expression of endophilin 1 (green traces), but not by expression of endophilin 1-ΔITSN (endophilin E329K + S336K mutant that does not bind ITSN-1) (pink traces). $N = 3$, 3 mice, TKO (25 cells), TKO + A1FL (30 cells) and TKO + A1 ΔITSN (35 cells). **g–m** Expression of endophilin 1 in endophilin TKO cells rescued exocytosis as inspected by combined capacitance and amperometry measurements (see Fig. 2), but the same effect could not be achieved by expressing an endophilin mutant that does not bind ITSN1 (E329K + S336K). Data shown in **g-m** are from four independent experiments, four mice each condition, cells TKO (39), TKO + A1-FL (36), and TKO + A1 ΔITSN (21). Scale bars 3 μm. Error bars denote SEM. $N$ = number of independent replicates. One-way ANOVA with Tukey's post-hoc test, \*$p < 0.05$, \*\*$p < 0.01$; \*\*\*$p < 0.001$, ns not significant.

endophilin mutant that does not bind intersectin-1 (indicated as endoA1-ΔITSN). The expression of endoA1-ΔITSN through the bicistronic lentiviral system was tested first in HEK-293 cells (Suppl. Fig. 7i). Remarkably, endophilin 1-ΔITSN was neither able to fully rescue exocytosis (Fig. 7g–j), including the burst and sustained component, nor the size and time constants of vesicle pools (Fig. 7k–m). Thus, endophilin's role in exocytosis is mediated, at least in part, through its interaction with intersectin-1.

**Intersectin's role in exocytosis is mediated via endophilin**. Similar to endophilin, ITSN-1 was redistributed in the chromaffin cells upon stimulation. Further, we found both ITSN-1 and intersectin-2 (ITSN-2) to be enriched on purified LDCVs (Suppl. Fig. 7g, h) and endophilin 1 and 2 colocalized significantly with ITSN-1 at the isolated plasma membranes (Suppl. Fig. 7k, note that both endophilins and intersectins have multiple interactors and functions at the plasma membrane, thus, a complete overlap of signals cannot be expected).

Like endophilin TKO chromaffin cells shown here, intersectin-1 KO chromaffin cells were reported to display exocytic defects[26,27]. To investigate the importance of the endophilin-intersectin interaction for chromaffin cell exocytosis, we took the complementary approach. We generated an ITSN1 mutant that does not bind endophilin 1 (ITSN1Δendo) by introducing two-point mutations (W949E and Y965E, according to Pechstein et al.[23]) in the SH3B domain of ITSN1, and verified the lack of interactions by a pull-down experiment (Fig. 8a). We then expressed either the ITSN1 WT protein, or the endophilin-binding deficient ITSN1Δendo mutant in the ITSN1 KO background using lentiviral transduction. The expression of ITSN1 WT and ITSN1Δendo via the bicistronic lentiviral system was comparable (Suppl. Fig. 7j). While the ITSN1 WT protein rescued secretion in ITSN1 KO chromaffin cells, the ITSN1-Δendo mutant was not able to fully rescue exocytosis to WT level (Fig. 8b–g). Specifically, burst release and sizes of vesicle pools were rescued by expressing ITSN1 WT, but not ITSN1Δendo (Fig. 8c, e), which mimics the phenotype of endophilin 1ΔITSN. The sustained component was not different between ITSN1 KO cells expressing ITSN1 WT and ITSN1Δendo (Fig. 8d). Expression of either ITSN1 WT or ITSN1Δendo did not alter the kinetics of the releasable pools (Fig. 8f, g). In sum, intersectin's role in exocytosis is mediated predominantly through its interaction with endophilin.

Overall, we observed that the exocytic defects in chromaffin cells without endophilin depend on endophilin's interaction with intersectin, while the exocytic defects in ITSN1 KO chromaffin cells depend on intersectin's interaction with endophilin. Therefore, we conclude that the functional intersectin-endophilin interaction coordinates exocytosis in chromaffin cells.

## Discussion

The first reports on endophilin linked its function to endocytosis[4,5,8,10,11]. Over 300 papers in the past 20 years built on these findings and helped to unveil the mechanisms of endophilin action in several types of endocytosis, including clathrin-mediated[8,10–12], clathrin-independent[15–17,59], and ultrafast endocytosis[18].

Our study shows that, in addition to its role in endocytosis, endophilin plays a key role in the priming and fusion of secretory vesicles. Endophilin is a peripheral protein with membrane-binding properties that appears to be present on some neurosecretory vesicles, as observed by a significant colocalization between the LDCV marker CgA and endophilin 1, or endophilin 2, respectively. Endophilin's role in exocytosis is mediated through its SH3-domain, and, specifically, through its SH3-domain

interactor intersectin-1, a partially membrane-associated protein that coordinates exocytosis and endocytosis and, like endophilin, translocates to the plasma membrane upon stimulation and regulates exocytosis[26,27].

**Endophilin's role in exocytosis is direct and endocytosis-independent**. We show that endophilin has a direct role in the vesicle exocytosis process itself: the expression of either endophilin 1 or endophilin 2 alone was sufficient to rescue all exocytic defects seen in neurosecretory cells without endophilin. Thus, the two endophilins have redundant roles in exocytosis. In addition, the expression of endophilin 1 or endophilin 2-BAR domain was not sufficient to produce a rescue of exocytosis in endophilin TKO cells. Next, endophilin that cannot bind intersectin-1 (E329K + S336K)[23] was not able to rescue intersectin's cellular distribution and exocytosis. None of the main exocytic proteins tested was found to be changed in the cells without endophilin, suggesting that the exocytic defect is not a result of altered protein sorting, or availability of exocytic machinery for secretion.

The role of endophilin in exocytosis appears to be independent of its well-established functions in endocytosis since the recycling/uptake of proteins (tested by transferrin and cholera toxin subunit-B uptake) and membrane (mCLING-Atto 647) were not majorly altered. We observed a mild decrease in the transferrin uptake efficiency between endophilin TKO and endophilin KOWTKO cells, yet no difference was detected between endophilin TKO and WT. This is an atypical finding since it suggests that endophilin KOWTKO cells were more efficient in the transferrin uptake than WT cells. Although we did not observe any significant changes in the total number of internalized vesicles with mCLING probe over the assayed time point in endophilin TKO cells, it is possible that the kinetics of the internalization, specifically in the first minutes, are altered—this could possibly affect the release site-clearance. However, a delay in release site-clearance cannot fully explain the observed phenotype, since chromaffin cells in culture do not have specialized fusion sites, and the RRP release is affected in endophilin absence. In addition, given that the LDCV generation/maturation steps take tens of minutes to hours, this short initial endocytic delay is likely not relevant for the protein recycling and generation of new LDCVs. In addition, we did not observe any overall differences in the levels and distributions of several additional vesicular proteins (e.g., VAMP2/synaptobrevin-2, synaptotagmin-1, synaptotagmin-7, etc.).

The magnitude of exocytic defects in endophilin TKO cells is not as striking as the loss of the main SNAREs, Munc13 and Munc18 (the key components of the exocytic machinery), or the simultaneous loss of both major calcium-sensors synaptotagmin-1 and −7[44,45,47,50,60]. Rather, the effect is intermediate, pointing to a regulatory role of endophilin in exocytosis.

**Endophilin's role in exocytosis is mediated via its SH3-domain and intersectin interaction**. We followed several leads that could explain endophilin's role in exocytosis. The absence of endophilin's SH3-domain reduced exocytosis even further than seen in endophilin TKO chromaffin cells, revealing a dominant-negative effect of the BAR domain alone, likely due to altered endophilin interactions and/or a competition with other peripheral vesicle proteins. Interestingly, out of all exocytic and endocytic (except clathrin) proteins inspected, only the distributions of intersectin-1 and intersectin-2 were altered. It was reported previously that the endophilin-1: intersectin-1 interaction is mediated through the SH3-domains of both proteins[23]. We now found that an endophilin-1 mutant that cannot bind intersectin-1 (E329K + S336K) was neither able to rescue the altered intersectin-1

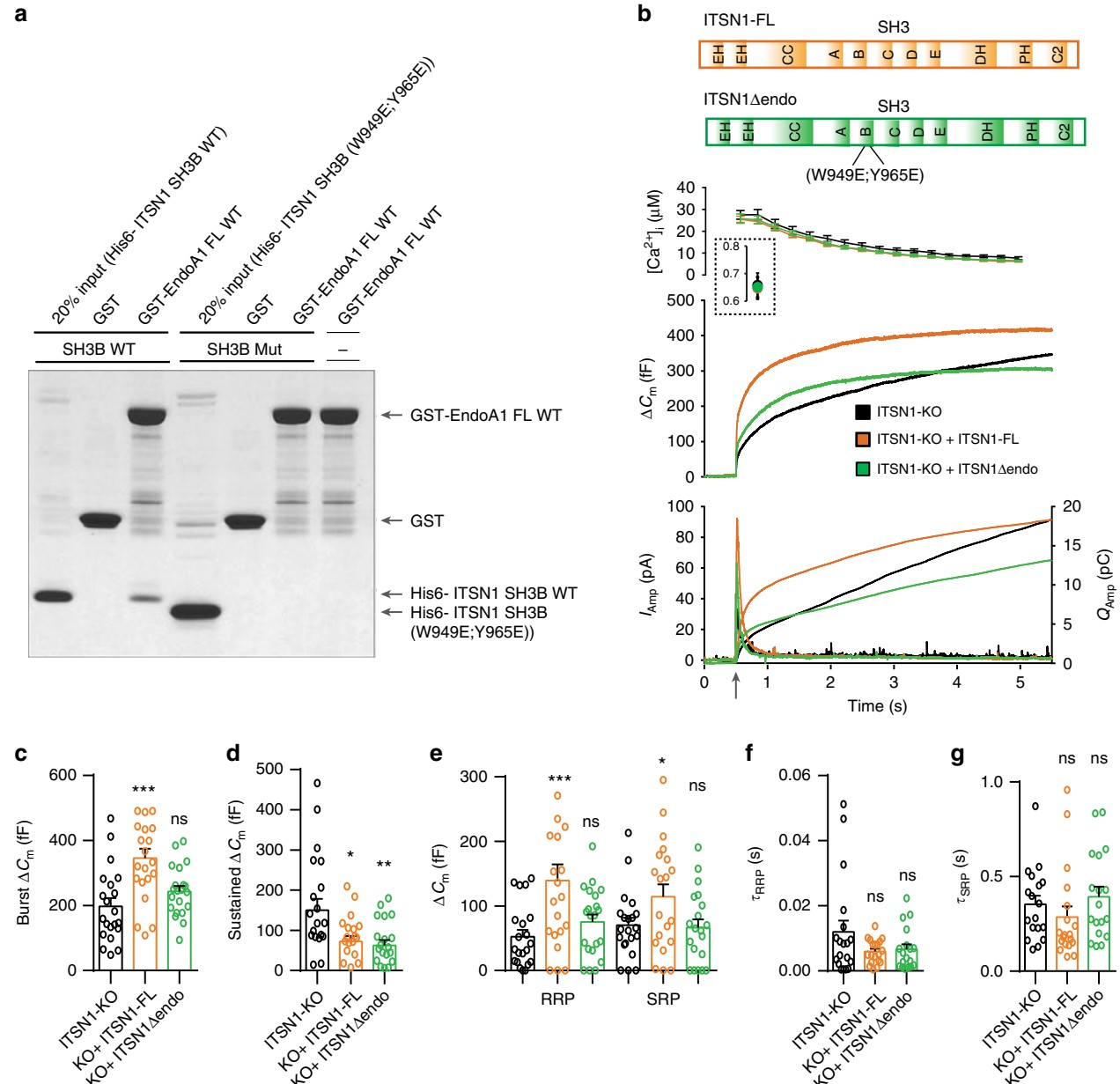

**Fig. 8 Intersectin's role in exocytosis is mediated through its interaction with endophilin. a** Pull-down experiment revealed an interaction between the SH3 domains of endophilin 1 and ITSN-1 that was perturbed when recombinant ITSN-1 with two-point mutations (W949 + Y965E) was used. Representative blot (three independent experiments were performed). **b** Exocytosis measured in ITSN1 KO cells (black traces) and ITSN1 KO cells expressing full-length proteins ITSN1 WT (orange traces) or ITSN1Δendo (W949 + Y965E in the SH3 domain, indicated in the schematic) (green traces). Panels are arranged similar to Fig. 1i. Note that ITSN KO cells expressing ITSN1 W949 + Y965E did not on average differ from ITSN1 KO cells (except for the sustained release), while expression of full-length ITSN1 WT was significantly higher. **c–g** Exponential fitting analysis revealed that burst (**c**), as well as the vesicle pool sizes (**e**), and kinetics (**f**, **g**), could be rescued by expression of ITSN1 WT, but no ITSN1 W949 + Y965E (sustained release could not be rescued). **b–g** $N = 3$, 3 mice each, number of analyzed cells: ITSN1-KO (18 cells), ITSN1-KO + ITSN-FL (19 cells), and ITSN1-KO + ITSN-Δendo (18 cells). Error bars denote SEM. $N =$ number of independent replicates. One-way ANOVA with Tukey's post-hoc test $*p < 0.05$; $**p < 0.01$; $***p < 0.001$, ns not significant.

distribution, nor the exocytic defect of endophilin TKO chromaffin cells. Consistently, in the complementary experiment with intersectin-1 KO chromaffin cells, the reduced exocytosis could not be rescued with an intersectin-1 mutant that cannot bind endophilin-1 (W949 + Y965E).

Both intersectin-1 and intersectin-2 were initially discovered as scaffold proteins involved in endocytosis[29,61–66]. Yet, further investigations showed that intersectin-1 is also implicated in

several other processes including exocytosis[26,28–30], thereby suggesting a role for intersectin in the coupling of exocytosis and endocytosis[24,25,30]. Intersectin-1 has an additional role in modulating actin dynamics in chromaffin cells, which might control the access of secretory vesicles to the plasma membrane[67], although recent examination of the Munc18-1-dependent regulation of the actin barrier shows that a strong actin barrier does not necessarily affect vesicle priming and fusion[67]. Actin also plays a

role in release site-clearance for vesicle docking and fusion[68]. The polymerization of the actin barrier seems primarily RhoA-dependent, while the de novo synthesis of actin is Cdc42-dependent[26,28,69,70]. We observed that, in the absence of endophilins, the actin barrier looks depolymerized in the resting condition, while stimulation increased actin's density (Suppl. Fig. 8) Thus, lack of endophilins led to a modulation of the actin barrier and the de novo synthesis. Therefore, the endophilin:intersectin interaction could also be a mechanism to control the actin barrier in chromaffin cells. Once the actin barrier is depolymerized at the specific sites, secretory vesicles can be recruited to their site of release at the plasma membrane, possibly through an interaction between intersectin-1 and the SNARE-protein SNAP-25 or SNAP-23[24].

We found that, without endophilin, intersectin mislocalized to the plasma membrane, and exocytosis was diminished. Strikingly, chromaffin cells without intersectin-1 also showed reduced exocytosis[27], which could be rescued by expression of full-length protein, but not an endophilin-binding mutant in our studies. In this context, endophilin may also be viewed as a regulator of intersectin in exocytosis and the direct interaction between endophilin and intersectin may coordinate the secretory vesicle priming and fusion.

**A putative model of endophilin's role in the regulation of exocytosis.** This study, along with our previous results on endophilins and intersectin-1[12,23] and the reported interaction between intersectin-1 and the SNARE proteins SNAP-25 and SNAP-23[24], as well as with the numerous intersectin studies (see above), support the model depicted in Suppl. Fig. 9. Whereas previous studies showed that the exocytotic proteins Munc18-1, synaptotagmin-1, SNAP-25 and syntaxin-1 are necessary for vesicle docking[60,71,72], endophilin and intersectin act in tandem to stimulate the priming and fusion of docked vesicles, likely through intersectin's interaction with SNARE proteins and/or a role in regulating the local actin network near the plasma membrane. Endophilin 1 and 2 as well as intersectin-1 are present on the majority but not all neurosecretory vesicles, and the effects of their absence are intermediate (when compared to SNAREs, Munc18, synaptotagmin-1 and -7), suggesting a regulatory role.

Once recruited to (the proximity of) the plasma membrane, we propose that endophilin and intersectin do not dissociate from the plasma membrane but take part in the endocytic process that follows the fusion of secretory vesicles. In addition to a complete LDCV fusion, kiss-and-run fusion is not uncommon in chromaffin cells[73]. Endophilin, as one of the main dynamin recruiters to the vesicle fission site, is therefore ideally positioned, together with intersectin, to act as a scaffold that couples vesicle fusion (exocytosis) and fission (endocytosis) events. This premise is supported by both endophilin's and intersectin's direct interactions with dynamin[4,24], as well as two studies in invertebrates: Bai et al.[19] suggested that endophilin is delivered to endocytic zones by exocytosis in *C. elegans*, and Winther et al.[74] showed that *D. melanogaster*'s Dap160/intersectin mutants lacking dynamin-binding do not properly accumulate dynamin in the periactive zone. Notably, we show that, without endophilin, intersectin is mislocalized; since endophilin can regulate intersectin's localization, this protein may be part of a check-point mechanism to ensure that intersectin acts only at the optimal time. Further studies are needed to understand whether this model, based on data from neuroendocrine cells, also can account for endophilin (and intersectin)'s roles in exocytosis at neuronal synapses.

In conclusion, our study suggests that the exocytosis of neurosecretory vesicles is regulated by endophilin 1 and endophilin 2, that were so far considered primarily as endocytic adaptors. This function of endophilin in exocytosis is dependent on intersectin: together they coordinate, at least in part, secretory vesicle priming and fusion.

## Methods

**Experimental models.** All animal-related procedures were performed according to the European guidelines for animal welfare (2010/63/EU), with the explicit permissions from the Niedersächsisches Landesamt für Verbraucherschutz und Lebensmittelsicherheit (LAVES), registration 14/1701, or the Danish Animal Experiments Inspectorate (2012-15-2935-00001 and 2018-15-0202-00157). Animals were housed and bred in the Zentrale Tierexperimentelle Einrichtung (ZTE) Göttingen, or at the AAALAC-accredited stable at the Department of Experimental Medicine, University of Copenhagen, with ad libitum access to water and food. Mice were kept in groups of 1–3 animals on a 12 h light/12 h dark cycle in individually ventilated cages. Endophilin mutant mice were originally generated and described in Milosevic et al.[12], and are accessible from the Jackson Laboratory (strain 021573—B6;129-Sh3gl2[tm1Pdc]/J; strain 021574—B6;129-Sh3gl1[tm1Pdc]/J; 021575—B6;129-Sh3gl3[tm1.1Itl]/J). ITSN-1 knock-out mice were originally described in Yu et al.[27]. Both male and female P0 pups were used to prepare cultures of adrenal chromaffin cells as follows: adrenal glands were isolated from P0 mice and placed in Locke's solution (154 mM NaCl, 5.6 mM KCl, 0.85 mM NaH$_2$PO$_4$, 2.15 mM Na$_2$HPO$_4$, and 10 mM glucose, pH 7.2), and connective tissue was cleaned away. The glands were digested in 200 μl enzyme solution (22.5 units/ml papain in Dulbecco's modified Eagle medium (DMEM) medium with high glucose (Gibco) supplemented with 25 mg L-cysteine, 125 μl 1 M CaCl$_2$, 1.25 ml 50 mM EDTA pH 7.4) at 37 °C for 35 min. The papain was inactivated by adding 200 μL inactivating solution (312.5 mg albumin fraction-V, 312.5 mg trypsin inhibitor, 112.5 ml DMEM medium with high glucose supplemented with 12.5 ml heat-inactivated fetal calf serum (Life Technologies)) for 5 min at 37 °C. This solution was replaced by 180 μL enriched DMEM medium (6.7 g DMEM powder with high glucose (Gibco) was dissolved in 496.5 ml double-distilled water supplemented with 0.55 g NaHCO$_3$, 1 ml penicillin/streptomycin (Life Technologies) and 2.5 ml insulin-transferrin-selenium-X (Thermo Fischer Scientific) and the glands were triturated (5–6 times) using a 200 μL pipette. The cells were plated on cleaned and sterilized glass coverslips and allowed to settle for 30–45 min before adding 1 ml enriched DMEM medium. The cells were maintained at 37 °C and 8 % CO$_2$, and used within 3–4 days for the experiments.

**Cloning and virus production.** For the rescue experiments, full-length endophilin 1, and endophilin 2, were cloned into the lentivirus (LV) expression vector FUGW (a gift of Oliver Schlüter, European Neuroscience Institute Göttingen, Germany) containing an IRES followed by an enhanced green fluorescent protein (EGFP) to allow simultaneous yet independent expression of both proteins. Endophilin 1 and endophilin 2 were first amplified by a PCR reaction (original plasmids were described in ref. [12]) and then inserted into the FUGW vector using *XbaI* and *BamHI* restriction enzymes. Similarly, endophilin 1-BAR and endophilin 2-BAR constructs (BAR domain and the linker sequence) were cloned by amplifying and inserting the endophilin 1-BAR and 2-BAR sequences into FUGW vector using *XbaI* and *BamHI* restriction enzymes. Endophilin 1ΔITSN (endophilin 1 E329K + S336K—mutant that cannot bind intersectin-1[23]) was first generated by Quik-Change II Site-Directed Mutagenesis Kit (Agilent) and subsequently inserted into the FUGW vector using *XbaI* and *BamHI* restriction enzymes. Intersectin-1 together with GFP was first extracted using *NheI* and *SalI* restriction enzyme (source plasmid Addgene #47395) and then inserted into the lentiviral vector (p156rrl-Syt1-SEP) using *XbaI* and *SalI* restriction enzymes. Intersectin-1Δendo (mutant intersectin-1 W949E + Y965E that cannot bind endophilin[23]) was generated by QuikChange II Site-Directed Mutagenesis Kit (Agilent) from the above described intersectin-1 in viral expression vector. All constructs were verified by sequencing and control restriction digestion. Constructs encoding the human intersectin-1-SH3B (aa 914-970) cloned in pET28a and recombinant rat endophilin A1 FL cloned into pGEX4T-1 (Amersham Biosciences) were published in Pechstein et al.[23].

Lentiviral particles were generated as follows: $1 \times 10^7$ HEK293FT cells were plated per Ø10cm dish. The cells were transfected with lentivirus transfer plasmid as detailed above (third generation lentivirus system) along with envelop and packaging plasmids using Lipofectamine-2000 and following the manufacturer's protocol (Invitrogen). The cells were maintained in the S2 bio-safety laboratory henceforth, and the medium was exchanged 14 h post-transfection. The medium containing lentivirus suspension was collected, centrifuged at 3000 RPM for 15 min at 4 °C to remove cell debris. Further, virus was concentrated using Amicon (100 K, UFC910096) at 4000 RPM for 20 min at 4 °C. The concentrated particles were diluted in Tris-buffer saline (TBS; pH 7.4); aliquots were frozen in cryo-tubes in liquid nitrogen and stored in −80 °C until being used. The efficiency of the lentivirus was tested by western blot and by imaging the intensity of the fluorescent reporter. The virus particles were added 6–8 h after chromaffin cell plating, and the cells were used 60–72 h post infection.

Lentiviral expression systems were verified in HEK-293 cells by western blotting and/or in chromaffin cells by measuring the fluorescence intensities of EGFP

expressed through bicistronic system. In either case, three independent experiments were performed, and each time new set of HEK-293 cells were transfected as indicated, collected, then proteins were extracted, quantified and inspected by western blot, as detailed below.

**Protein expression, purification, and pull-down.** Recombinant human intersectin-1 SH3B (aa 914-970) and recombinant rat endophilin A1 FL were expressed by *E.coli* in 2xYT medium (Sigma-Aldrich) overnight at 18 °C (induction at $OD_{600}$ 0.5-0.7 with 1 mM isopropylthio-β-galactoside, IPTG). Bacterial cells were collected by centrifugation (6000 x *g*, 4 °C, 10 min), lysed and supernatant was collected after the second centrifugation (15,000 x *g*, 4 °C, 20 min). The GST and His-fusion proteins were then purified by affinity chromatography using commercial GST-Bind resin (Novagen) and His-Select Nickel Affinity Gel (Sigma) following the manufacturer's protocols, and subsequently subjected to gel filtration on a Superdex 75 16/60 column (GE Healthcare) buffered in phosphate buffered saline (PBS), pH 7.4. The affinity tags were removed by incubation with thrombin (1 U/mg of protein, 1 h, 37 °C) before gel filtration. The purified proteins were used for the pull-down experiments. In short, 50 or 100 µg GST-fusion proteins were coupled to GST-Bind resin (Novagen) and incubated with 10 µg $His_6$-tagged recombinant proteins, respectively, in a total volume of 1 ml for 1 h at 4 °C on a rotating wheel. Following extensive washes, samples were eluted and analyzed by direct Coomassie staining and/or sodium dodecyl sulfate–polyacrylamide gel electrophoresis (SDS-PAGE) and immunoblotting.

**Western blotting.** Standard SDS-PAGE blot was used to analyze total protein levels. An electrophoresis system (BIO-RAD) was used to perform the separation using custom-prepared gels (10% or 12%, pH 8.8), depending on the size of a protein to be analyzed by immunoblotting. Protein samples were prepared from intact adrenal glands (note that glands from 3–4 animals of the same genotype were pooled together, and referred to as one biological sample) in RIPA buffer (50 mM Tris pH 8.0, 150 mM NaCl, 1% Triton X-100, 0.1% SDS and 0.5% sodium deoxycholate, SDS, with fresh 1x protease/phosphatase inhibitor supplement, Roche), mixed with 6x Laemmli buffer (4.16 M SDS, 47 ml glycerol, 0.9 mM bromophenol blue, 12 ml 0.5 M Tris pH 6.8, 5 µl β-mercaptoethanol in 100 ml $dH_2O$) and denatured for 5 min at 95 °C. Forty micrograms of sample/well was loaded. After electrophoresis, the proteins were transferred onto a nitrocellulose membrane using the transfer system (BIO-RAD). The membranes were further blocked in 5% milk prepared in 1x Tris-Buffered Saline (TBS) and 0.1% Tween 20 (blocking buffer) at room temperature for 1 h, and subsequently incubated with primary and secondary antibodies (diluted in the blocking buffer). The proteins were detected using the Odyssey infrared imaging system (Li-COR) and analyzed using Image Studio Lite (a software package from LI-COR Biosciences) and/or ImageJ (http://rsb.info.nih.gov/ij/index.html). Both software were used to compare the density (i.e., intensity) of bands on a digital image of the western blot after subtracting the background signal from the adjacent area. original western blot data are shown in Suppl. Fig. 10.

**Electrophysiological measurements.** The mouse chromaffin cells were maintained in extracellular solution (145 mM NaCl, 2.8 mM KCl, 2 mM $CaCl_2$, 1 mM $MgCl_2$, 10 mM HEPES, and 2 mg/ml D-glucose, pH 7.2, 305 mOsm/kg) during the electrophysiological recordings, which were performed at room temperature (22–24 °C). Capacitance and amperometric measurements were performed in parallel on a Zeiss Axiovert 10 equipped with Polychrome V monochromator (Till Photonics), an EPC-9 amplifier (HEKA Elektronik) for patch-clamp capacitance and an EPC-7 (HEKA Elektronik) for amperometry. Catecholamine release was triggered by ultraviolet-flash photolysis (using a JML-C2, Rapp Optoelektronik) of a caged calcium compound, nitrophenyl-EGTA, which was transferred into the cell through a patch pipette. The setup calibration was done by infusion of eight solutions of known calcium concentrations into chromaffin cells, by ratiometric measurement of two calcium dyes with different calcium binding affinity, Fura4F and Furaptra (ThermoFischer Scientific) and this allowed intracellular calcium monitoring. The excitation light (Polychrome V) was alternated between 350 and 380 nm to perform the ratiometric measurements of $[Ca^{2+}]_i$. The emitted fluorescence was detected with a photodiode and sampled using Pulse software (HEKA). The same software was used to control the voltage in the pipette and perform capacitance measurements. The intracellular solution contained (in mM): 100 Cs-glutamate, 8 NaCl, 4 $CaCl_2$, 32 Cs-HEPES, 2 Mg-ATP, 0.3 GTP, 5 NPE, 0.4 Fura4F, and 0.4 Furaptra (L-ascorbic acid was added to prevent flash-induced damage to Fura dyes), pH 7.2 (osmolarity adjusted to ~295 mOsm/kg).

Amperometric recordings were done using Ø5 µm carbon fibers (Thornel P-650/42, Cytec) insulated by the polyethylene method[75] and EPC-7 (HEKA Elektronik). Cells were perfused with intracellular solution for single amperometry spikes, consisting of 70 mM Cs-glutamate, 8 mM NaCl, 4 mM $CaCl_2$, 22.5 mM Cs-HEPES, 2 mM Mg-ATP, 0.3 mM Na-GTP, 37 mM $Ca^{2+}$ DPTA, 0.32 mM Fura-4F and 0.48 mM Furaptra pH 7.2 (osmolarity adjusted to ~300 mOsm/kg). Fibers were clamped to 700 mV. Currents were acquired at 25 kHz and filtered off-line using a Gaussian filter with a cutoff set at 1 kHz.

Electrophysiological and amperometry experiments were performed in such way that conditions could be compared side-by-side: e.g., the WT and mutant constructs are expressed in chromaffin cells from the same animal (e.g., TKO cells), and the same number of cells for each condition is recorded on each experimental day. If comparing various genotypes (e.g., KOWTKO and TKO), this is done exclusively in littermates, prepared in parallel and recorded on the same day. Number of cells indicated in Figure legends represent biological replicates; cells are isolated from several animals (as indicated), and at least three sets of recordings from independent cell preparations from 1 or 2 litters (if two litters are born on the same day) were done for each dataset, often more (as indicated).

**Plasma membrane sheet assay.** Plasma membrane sheets were generated from cultured mouse chromaffin cells by adapting a protocol originally described in Milosevic et al.[39] for (large) bovine chromaffin cells. Briefly, the cells were subjected to a brief ultrasonic pulse in the sonication buffer (120 mM potassium glutamate, 20 mM potassium acetate, 20 mM HEPES, 4 mM $MgCl_2$, 4 mM EGTA, 6 mM $Ca^{2+}$-EGTA, pH 7.3, 300 mOsm/kg) in order to generate thin, flat inside-out plasma membrane sheets. Detailed protocol for mouse chromaffin plasma membrane sheets generation is described elsewhere[40]. The plasma membrane sheets were immediately fixed with 4% freshly prepared paraformaldehyde (PFA) in PBS (Sigma) for 1 h before proceeding with the immunostaining procedure as detailed below.

**Immunocytochemistry.** Chromaffin cells were cultured for up to 72 h on poly-L-lysine coated coverslips in a 12-well plate (Sarstedt) for immunocytochemistry experiments. The cells were fixed in freshly prepared 4% PFA in PBS and neutralized afterwards for 20 min with 50 mM $NH_4Cl$ in PBS. Blocking was performed with a blocking buffer containing 3% bovine serum albumin (BSA; Sigma), 0.2% cold-fish gelatine (Sigma) and 1% goat serum (Gibco) for 1 h, and cells were permeabilized in 0.1% Triton X-100 (Sigma) in the blocking buffer for 10 min. After a brief washing step, the cells were stained with primary antibody overnight at 4 °C followed by washing and secondary antibody staining in the dark at room temperature for 1 h (the antibodies are listed in Key Resources Table; the endophilin antibodies were characterized as specific, since little to no signal was detected in TKO cells). The washing procedure was repeated following 2 min incubation with 4′,6-diamidino-2-phenylindole (DAPI 1:5000 in PBS; Sigma) to stain the nucleus of the chromaffin cells. Finally, the coverslips were mounted in Mowiol 4-88® mounting medium (Sigma). Images were acquired using Zeiss LSM 710 laser scanning confocal microscope or Zeiss LSM 800 Airyscan confocal microscope (63x objective, numerical aperture 1.4).

For chromaffin cell stimulation, the cells were washed carefully in pre-warmed Locke's solution before incubation in extracellular (control condition) or high $K^+$ solution (88 mM NaCl, 59 mM KCl, 2 mM $CaCl_2$, 1 mM $MgCl_2$, 10 mM HEPES, and 2 mg/ml D-glucose, pH 7.20, 300 mOsm/kg) for 3 min at room temperature (RT). The cells were placed on ice immediately, fixed in 4% PFA (freshly prepared in PBS) for 10 min on ice followed by 20 min at RT.

**Electron microscopy on adrenal mouse chromaffin cells.** Adrenal glands from endophilin TKO, endophilin KOWTKO and WT P0 mice were isolated and subsequently fixed in 4% PFA + 0,5% glutaraldehyde (GA; Sigma) in 0.1 M PBS, pH 7.2 for 1 h on ice, and afterwards in 2% GA in 0.1 M sodium cacodylate buffer (pH 7.2; Sigma) overnight at 4 °C. The next day, the adrenal glands were washed three times for 10 min in 0.1 M sodium cacodylate buffer. Post-fixation was done on ice for 1 h in 1% (v/v) $OsO_4$ in cacodylate buffer, followed by further washing steps (2 × 10 min cacodylate buffer, 3 × 5 min water). En-bloc staining of adrenal glands was performed using 1% (v/v) uranyl acetate (Sigma) in water for 1 h on ice. Subsequent to three brief washing steps in water, adrenal glands were dehydrated in an ascending ethanol series (5 min 30%, 5 min 50%, 10 min 70%, 2 × 10 min 95%, 3 × 12 min 99.9% ethanol) and infiltrated in Epon resin (50% ethanol + 50% epon for 30 min and for 90 min, 100% epon for ~20 h) at room temperature. The samples were placed in the embedding molds and polymerized for 48 h at 70 °C. Ultrathin sections (65 nm) were cut using a Leica UC6 ultramicrotome, placed on formvar-coated copper grids, and stained in uranyl acetate and lead citrate (using the Reynold's method). Images were acquired as detailed in Kroll et al.[76] by a JEOL JEM-1011 transmission electron microscope (on average 5 images/cell were acquired).

Dissociated chromaffin cell cultures were prepared for ultrastructural analysis as in[49]. Briefly, the cells were fixed with 2.5% glutaraldehyde in 0.1 M cacodylate buffer (pH 7.4) for 1 h at RT, followed by overnight at 4 °C. Cells were washed after fixation and post-fixed for 1 h at RT with 1%$OsO_4$/1%$KRu(CN)_6$. After dehydration through a series of increasing ethanol concentrations, cells were infiltrated with Epon resin that was polymerized at 65 °C and the coverslip was removed by alternatively dipping it in hot water and liquid nitrogen. Regions with high density of chromaffin cells were selected by observing flat Epon-embedded cell monolayer under the light microscope and mounted on Epon blocks for thin sectioning. Ultrathin sections (80 nm) were cut parallel to the cell monolayer on an Ultracut ultramicrotome (Leica), collected on single-slot, Formvar-coated copper grids and contrasted by 0.5% uranyl acetate and Reynold's lead citrate in an AC20

ultrastainer (Leica). Images of chromaffin cell cross sections were taken on a JEOL1010 TEM with a Modera side-mounted CCD camera (EMSIS) at 60 kV at 10k (overview) and 30k (membrane area) fold magnification.

**Endocytosis assays on chromaffin cells.** Transferrin (Tf) was conjugated to Alexa Fluor[TM] 546 (5 μg/ml; Invitrogen, Cat# T23364), and the uptake assay in adrenal chromaffin cells was performed as in Chen et al.[77]. All images were captured under the same acquisition settings using a Zeiss 800 Airyscan confocal microscope. Analysis of Tf-A546 data (z-stack of whole cells) was performed by Imaris (Bitplane) using the Spot module. Non-toxic recombinant cholera toxin subunit-B (CT-B) was conjugated to Alexa Fluor[TM] 594 (Thermo Fischer, Cat# C22842), and the uptake assay was adapted to adrenal chromaffin cells using the protocol from Kirkham et al.[78] In short, uptake of 2 μg/ml CT-B-A594 was carried out at 37 °C in serum-free medium (Gibco) for 8 min (note that CT-B-A594 attaches to cells by binding to ganglioside $GM_1$), Cells were washed $4 \times 30$ s with the extracellular buffer to remove CT-B-A594 cell surface labeling, fixed in 4% PFA for 20 min, and imaged with a Zeiss 800 confocal microscope. Images were captured through the equatorial plane of each cell (one plane only) using the same acquisition settings and were analyzed in ImageJ (a background-threshold was applied and every fluorescent cluster greater than six pixels was counted).

Live imaging of chromaffin cells labeled with mCLING-Atto647 (Synaptic Systems) was done using the spinning disk confocal microscope (Perkin Elmer/Nikon/Velocity) with temperature control unit (kept at 37 °C) and custom-built imaging chamber. Cells were maintained in extracellular solution in imaging chamber and stained with 0.5 nmol/ml mCLING-Atto647 for 1 min. The solution was exchanged and cells were washed rapidly (few seconds) before image acquisition. Images were captured up to 8 min after addition of mCLING-Atto647 through the middle of each cell (using the same acquisition settings) and quantified using ImageJ as detailed above for CT-B-A594.

**Quantification and statistical analysis.** Unless otherwise stated, in the figure legends, N represents number of independent experiments, and n denotes individual cells/biological samples obtained from N (or more) animals. Colocalization analysis in mouse chromaffin cells was performed as follows: colocalization was evaluated using object-based overlap and JACoP plugin in ImageJ since endophilin signals had a cytosolic component. Specifically, a region-of-interest (ROI) was chosen so it did not contain nucleus, and the images were segmented into objects and background (bright fluorescent objects were segmented from the image) before Pearson's correlation coefficient was calculated. The same analysis was performed with the 90° rotated image, and this random value was subtracted for each cell. In addition, a complementary manual colocalization analysis was performed: here, circles were superimposed on bright fluorescent spots (e.g., in the CgA channel) and transferred to identical image locations in the endophilin channel. If the fluorescence intensity maximum in the endophilin channel was located in the same circle and the morphology of the signal resembled that of the CgA signal, the circle was rated as positive (colocalized). If this was not the case, it was rated as negative (not colocalized). To correct for random colocalization of two abundant signals, circles were also transferred to a 90° rotated image of the endophilin channel. At least 9 images/experiment from 3 to 5 experiments were analyzed for each genotype/condition. The two approaches (semi-automatic ImageJ-based and manual) gave similar result.

Colocalization analysis on the plasma membrane sheets was performed as follows: a ROI was chosen on the plasma membrane in the blue channel (TMA-DPH staining) and transferred to the green (endophilins) and red (CgA or ITSN1) channels, and the mean cross-correlation coefficient was calculated between green and red signals using Igor Pro (Wave Metrics). The same analysis was performed with the 90° rotated image, and this random value was subtracted for each respective measurement. In addition, a complementary manual colocalization analysis was performed as detailed above; the two approaches gave similar result.

The kinetic analysis of the capacitance measurements was performed by fitting individual capacitance traces with a triple-exponential function using IGOR Pro software, as in Milosevic et al.[39]. The amplitudes and time constants of the two faster exponentials define the size and release kinetics of the slowly releasable pool (SRP) and the readily releasable pool (RRP), respectively. Filtering, spike detection and analysis of amperometric spikes were performed by a in IGOR Pro (Wave Metrics) as in ref. [79]. Data are represented as mean ± SEM, and unpaired two-sided t-test (for two datasets), and nonparametric Kruskal–Wallis test with Dunn's multiple comparison test or one-way ANOVA after Tukey's post-hoc test (for three datasets) were used to test statistical difference, which is indicated by *$p < 0.05$, **$p < 0.01$, and ***$p < 0.001$.

For the ultrastructural analysis by EM, independent embeddings per group were analyzed using IMOD (bio3d.colorado.edu/imod) and ImageJ software (analysis was blind to experimental conditions). The area of chromaffin cells was defined as area within the plasma membrane, thereby excluding the nucleus area. The area of LDCVs was directly measured using the area selection tool in ImageJ. The statistics is done by one-way ANOVA after Tukey's post-hoc test (Suppl. Fig. 3b–e) and Kolmogorov–Smirnov test (Suppl. Fig. 3f, g; to ensure that the detected differences were not artefacts created by multiple comparisons, we applied Bonferroni-correction). For the cultured cells analysis, quantitative morphometric analysis was performed in iTEM software (EMSIS) blinded for experimental conditions. The

data obtained from individual cells was nested within individual animal (more than one observation drawn from one independent sample, an animal). To accommodate potential dependency, we tested the effects by multilevel analysis using SPSS (IBM) software[80].

For quantification of CgA-positive vesicles (Fig. 4a, b) and Syt-1 intensity on CgA-positive vesicles (Fig. 4c, e), three-dimensional surface reconstruction and immunofluorescence signal analysis was carried out with the "Cell with organelles" module of Imaris software, version 8.0.2 (Bitplane).

For quantification of intensity of proteins detected by immunofluorescence (Fig. 4d, g and Fig. 5a), the mean fluorescence intensity was measured in the cell cytoplasm, excluding the nucleus, using ImageJ and normalized to the measured area. The data are represented as mean ± SEM and statistics was performed by unpaired t-test.

To characterize the redistribution of endophilins and intersectins (Figs. 1e–h and 7a–f; Suppl. Fig. 7c–f), we capitalized on the round-shape of adrenal chromaffin cells and defined a ROI-based analysis approach in ImageJ software. Two concentric ROIs were defined: the outer circular ROI around the whole cell and the inner circular ROI (to measure the intensity of the cytosol). The inner circular ROI was defined to be 40 pixels less than the radius of the outer ROI. The intensities of two ROIs were then measured, and the inner circular ROI was subtracted from the outer circular ROI to calculate the near-membrane intensity. The ratio of intensities between membrane and cytosol was plotted. The data are represented as mean ± SEM.

**Reporting summary.** Further information on research design is available in the Nature Research Reporting Summary linked to this article.

## Data availability
All key datasets generated and analyzed during this study are included in this manuscript and Supplementary Data. The raw datasets are available from the corresponding authors on reasonable request.

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

## Acknowledgements

We thank the UMG animal facility, M. König and M. Costa for the excellent assistance and for help with genotyping, A. Witkowska for purified LDCVs, B. Tawfik for Syt1-pHluorin and Syt7-pHluorin constructs, and P. De Camilli (Yale University) for endophilin KO mice and reagents. This work was supported by Schram-Stiftung T287/25457 and Deutsche Forschungsgemeinschaft (Emmy Noether Young Investigator Award MI-1702/1) to I.M., SySy fellowship to S.G., the Lundbeck foundation (P.S.P., S.H., J.B.S.), ERC Starting Grant 337327 (N.R.), ZonMW 91111009 (J.R.T.v.W.), Alzheimer's Associaton AARG-17498856 (J.R.T.v.W.), the Novo Nordisk Foundation (J.B.S.) and the Independent Research Fund Denmark (S.H., J.B.S.).

## Author contributions

Conceptualization: I.M.; experiments and/or analysis: S.G., I.M., V.S., S.H., J.P.C., J.K., P.S.P., M.G., N.H., A.P., N.S., N.R., T.M., and J.R.T.v.; reagents: D.S., V.H., J.B.S., and I.M.; supervision: V.H., J.B.S., and I.M.; writing and revision: S.G., J.B.S., and I.M. with input from all coauthors.

## Competing interests

The authors declare no competing interests.
