## [Peer Review File · Nature Communications]

Reviewers' Comments:

Reviewer #2:

Remarks to the Author:

The authors went through substantial efforts of revising the manuscript. They have done well in characterizing the general status of the exocytotic machinery in the TKO, and now provide more data concerning the functional relation to intersectin.

Their principal observation is new and surprising. Though the actual mechanistic rooting of this phenotyp remains to be explained, the manuscript in its current form appears ready for publication. I unreservedly suggest publication in Nature Communication.

Reviewer #3:

Remarks to the Author:

The authors have fully addressed my concerns. I do not have any further suggestions.

My comments on Reviewer 1's concerns from the previous round:

Major comment 2. The authors found that endophilin TKO cells expressing synaptotagmin-1-pHluorin show a lower fluorescence increase upon stimulation than WT and littermate controls (new Suppl. Fig. S2J-L). The authors concluded that this result thereby confirms that loss of endophilins in chromaffin cells impairs exocytosis.

I do not agree with the analysis method. I think the authors should count the number of spots induced by a stimulation rather than the overall fluorescence increase from the bottom of the cell. The number of pHluorin spot reflects the number of vesicles being released. A decrease of the spot number indicates a decrease of the released vesicle number. The results should be re-analyzed or redone.

Major comment 3. The authors did not perform the experiments the reviewer suggested. The reviewer suggested to recording the localisation of fluorescently tagged endophilin and/or intersectin during release events in time-lapse experiments using TIRF microscopy, which would strongly support the authors' model. Instead, the authors show colocalization of endophilin with vesicles at the plasma membrane sheets after cell unroofing. It is unclear whether these vesicles undergo exocytosis.

Major comment 4. The authors did not perform experiments the reviewer suggested. The authors explained the difficulty. They showed no statistical difference in expression of VAMP2 in Fig. S4E, but with significant variation that cannot exclude a 50% reduction of VAMP2. Such a reduction of VAMP2 expression could explain the exocytosis defect observed with endophilin triple knockout. At least the VAMP2 expression in Fig. S4E should be repeated to reduce the data variation.

Major comment 6. The authors did not perform experiments the reviewer suggested. Fig. S2K plots the fluorescence level but not the pHluorin spot number, which may not reflect the exocytosis

number. The reviewer hope to see that in the presence of pitstop, TKO still reduces the capacitance increase.

Reviewers' comments:

Reviewer #2 (Remarks to the Author):

The authors went through substantial efforts of revising the manuscript. They have done well in characterizing the general status of the exocytotic machinery in the TKO, and now provide more data concerning the functional relation to intersectin.

Their principal observation is new and surprising. Though the actual mechanistic rooting of this phenotyp remains to be explained, the manuscript in its current form appears ready for publication. I unreservedly suggest publication in Nature Communication.

Thank you!

Reviewer #3 (Remarks to the Author):

The authors have fully addressed my concerns. I do not have any further suggestions.

Thank you!

Comments on Reviewer 1's concerns from the previous round:

Major comment 2. The authors found that endophilin TKO cells expressing synaptotagmin-1-pHluorin show a lower fluorescence increase upon stimulation than WT and littermate controls (new Suppl. Fig. S2J-L). The authors concluded that this result thereby confirms that loss of endophilins in chromaffin cells impairs exocytosis.

I do not agree with the analysis method. I think the authors should count the number of spots induced by a stimulation rather than the overall fluorescence increase from the bottom of the cell. The number of pHluorin spot reflects the number of vesicles being released. A decrease of the spot number indicates a decrease of the released vesicle number. The results should be re-analyzed or redone.

pHluorin-based experiments are done rarely in adrenal chromaffin cells, in part since the combination of fast capacitance and amperometry measurements is generally accepted as the most comprehensive strategy to analyze chromaffin cell exocytosis with the necessary temporal resolution. Of note, amperometric current reports directly on adrenaline release, and capacitance measurements reveal the exocytotic capacity of the entire cell, whereas pHluorin assays are typically restricted to one optical plane (usually a cell footprint). Nonetheless, we followed Reviewer's suggestion to also do pHluorin-based experiments in chromaffin cells isolated from endophilin mutants.

We cloned and produced lentivirus encoding synaptotagmin-1-pHluorin and established the pHluorin assay in primary culture of adrenal chromaffin cells. The assay was verified under two stimulation conditions (nicotine and KCl), and in presence of the endocytosis inhibitor Pitstop-2. We originally adopted the same data analysis approach as used for pHluorin experiments in neuronal cells (Sankaranarayanan et al., Biophys J 2000 - 11023924; Milosevic

et al., Neuron 2011 - 22099461). In short, we quantified the increase in fluorescence (ΔF) upon cell stimulation, since the pHluorin moiety is responsive to changes in pH and the increase in fluorescence is correlated to vesicles being exocytosed and exposed to the neutral pH. Importantly, quantification of the total fluorescence is the simplest way to analyze these data, as this method does not depend on decisions on what constitutes a vesicle.

However, we see a value in the alternative approach suggested by the Reviewer. Thus, we performed the analysis as suggested, basically by following the method described by Rao *et al.*, J Gen Physiol 2017 – 28687607. Since synaptotagmin7-pHluorin (syt-7-pHluorin) seems to be a more suitable pHluorin probe for chromaffin cells (we realized this only after performing syt-1-pHluorin experiments - although started simultaneously the cloning and producing lentivirus expressing syt-7-pHluorin was challenging and took longer than syt-1-pHluorin) and it is predominantly used in the Rao *et al.* (2017) publication, we have performed experiments also with this construct in WT and endophilin mutant cells (importantly, both syt-1 and syt-7 play a role in chromaffin cell exocytosis).

We were happy to observe that both constructs, syt1-pHluorin and syt7-pHluorin, revealed the same result: endophilin TKO cells expressing syt-1-pHluorin, or endophilin TKO cells expressing syt-7-pHluorin, showed a lower number of fluorescent puncta upon stimulation than WT and littermate controls (new Suppl. Fig. S2), thereby confirming that loss of endophilins in chromaffin cells impairs exocytosis.

Finally, while the exocytic defects observed in endophilin TKO cells are substantial and confirmed by various methods, they are not as striking as the loss of SNAP-25, syntaxin-1 or Munc18-1 (the key components of the exocytic machinery), or the simultaneous loss of both major calcium-sensors synaptotagmin-1 and synaptotagmin-7. Endophilins are likely potent regulators of chromaffin cell exocytosis but they are not essential for secretion, as discussed in the revised manuscript.

Major comment 3. The authors did not perform the experiments the reviewer suggested. The reviewer suggested to recording the localisation of fluorescently tagged endophilin and/or intersectin during release events in time-lapse experiments using TIRF microscopy, which would strongly support the authors' model. Instead, the authors show colocalization of endophilin with vesicles at the plasma membrane sheets after cell unroofing. It is unclear whether these vesicles undergo exocytosis.

We would love to be able to do the TIRF microscopy-based experiments as suggested originally. Of note, we have attempted to do these experiments at two independent occasions, yet technical issues prevented further advances along this line for a number of reasons: (1) Mouse adrenal chromaffin cells are small round cells that do not adhere well to coated coverslips and form very small cell footprints ($\sim 3 \mu\text{m}$, or smaller) making it difficult to reliably follow the few exocytic events by TIRF microscopy. (2) In our hands, it is impossible to achieve efficient transfection of mouse chromaffin cells without virus. Both endophilins and intersectins are in part cytosolic proteins: when expressed through lentiviral- or Semliki Forest Virus-systems, the cytosolic pools of endophilins and intersectins are abundant and mask

endophilin/intersectin dynamics in the proximity of the plasma membrane. (3) To document endophilin/intersection specifically on the fusing vesicles it would be necessary to co-express fluorescently-labeled endophilin/intersectin with a pH-sensitive vesicle marker (e.g. syn-7-pHluorin) and follow them by dual-color imaging, which would necessitate the construction of novel viral vectors expressing both proteins, which is likely to fail due to the large size of the proteins involved (in our hands co-transfection with several viruses does not lead to consistent co-expression in most chromaffin cells). Of note, making lentiviral constructs that express only the large protein intersectin-1 (~170 kDa) took 2.5 months, with two laboratories working on it in parallel. (4) This approach would be based on the studies of overexpressed proteins, and not endogenous endophilin and intersectin. As much as we would like to address these questions by TIRF microscopy, which works nicely with other models and cells, this approach is unfortunately not feasible in this particular case. We very much hope the Reviewer concurs.

We have spent several months to address this question by other means. The most promising approach turned out to be the adaptation of the plasma membrane sheet technique to mouse chromaffin cells, which also allowed us to study the distribution and behavior of the endogenous proteins. Upon immediate fixation and immunostaining with anti-CgA (a marker of LDCVs) and anti-endophilin 1/anti-endophilin 2 specific antibodies, we found significant colocalization between endogenous CgA-positive LDCVs and both endogenous endophilin 1 and 2 on the plasma membranes (new Figure 1C-D). Since only docked/primed secretory vesicles are detectable with this approach, we are confident that at least some secretory vesicles contain endophilin 1 and 2. To address further concerns that these secretory vesicles are non-functional, we list some examples where the plasma membrane sheets with attached secretory vesicles have been used to study the principles of vesicle release on the plasma membrane: Avery *et al.*, J Cell Biol 2000 - PMID 10648564, Lang *et al.*, EMBO J 2001 – 11331586; Holroyd *et al.*, PNAS USA 2002 - 12486251; Lang *et al.*, J Cell Biol - 12177041, Dernick *et al.*, Nat Cell Biol 2003 - 12652310; Barszczewski *et al.*, Mol Biol Cell 2008 – 18094056, etc.. Further, to illustrate that secretory vesicles in our model are also capable of undergoing exocytosis, we include a figure below for the Reviewer's consideration.

Figure for Reviewer. Ca²⁺-dependent exocytosis of chromaffin LDCVs on the plasma membrane sheets, monitored by time-lapse microscopy using NPY-Venus as a content marker. Chromaffin membrane sheets with attached LDCVs labelled with NPY-Venus were pre-incubated in 'priming' buffer containing 3 mg/ml rat brain cytosol, 2 mM ATP and 300 nM [Ca²⁺]_{free}, in buffer containing 3 mg/ml rat brain cytosol and no free calcium and in 'priming' buffer with 2 μM BoNT/C (botulinum toxin type C inhibiting exocytosis by cleaving the SNARE protein syntaxin-1 was used as negative control) at 37°C.

To stimulate fusion, sheets were treated with 'fusion' buffer containing 3 mg/ml rat brain cytosol, 2 mM ATP and 30 μM [Ca²⁺]_{free}. 43 plasma membrane sheets with 984 labelled vesicles were analyzed in total. (A) 6 fluorescent vesicles imaged for 15 min. (B) Changes in fluorescence intensity of NPY-Venus labelled LDCVs: Exemplary traces of three vesicles shown in A. Intensity values were corrected for local background, normalized to initial intensity and plotted against time. Vesicles (1) did not change fluorescence intensity, (2) became brighter, presumably due to pH sensitivity of Venus fluorescent protein, or (3) disappeared. (C-D) Exocytosis is dependent on the presence of Ca²⁺. Exocytotic membrane fusion was calculated as the number of LDCVs scored positive for exocytosis divided by a number of LDCVs present before the 'fusion' buffer addition. Values are given as mean ± SEM.

We however consider that including this figure would not bring new information to the already data-rich manuscript, and may distract from the original message. Thus, we opted not to include this Figure into the revised manuscript; examples from the literature are cited instead.

Major comment 4. The authors did not perform experiments the reviewer suggested. The authors explained the difficulty. They showed no statistical difference in expression of VAMP2 in Fig. S4E, but with significant variation that cannot exclude a 50% reduction of VAMP2. Such a reduction of VAMP2 expression could explain the exocytosis defect observed with endophilin triple knockout. At least the VAMP2 expression in Fig. S4E should be repeated to reduce the data variation.

Following Reviewer's advice, we repeated the qPCR experiments for synaptobrevin-2/VAMP2 with 5 animal samples/condition (we have also repeated experiments with synaptotagmin 1 as well). The variability between samples is now lower, but the findings are the same - there is no detectable difference in the levels of VAMP2 and synaptotagmin 1 transcripts (new Suppl Fig S4E of the revised manuscript). Kindly note that detected VAMP2 transcript levels match very well the results of Western blots that inspected the protein levels of VAMP2 (Suppl Fig S4C - quantified in S4D, lower panel).

Of note, in mouse chromaffin cells Ca^{2+} -triggered secretion is only moderately altered in the complete absence of synaptobrevin-2/VAMP2 (VAMP2 KO), likely due to compensation by cellubrevin (Borisovska et al. EMBO J 2005 - PMID: 15920476). Thus, it is unlikely that a reduction in synaptobrevin-2/VAMP2 expression, even if it were detected in this model system (but we did not detect it), would explain the exocytic defect observed in endophilin TKO cells.

Besides synaptobrevin-2/VAMP2, we have comprehensively tested the transcript levels and/or protein levels for 18 proteins associated with chromaffin cells secretion: chromogranin-A, chromogranin-B, synaptotagmin 1, synaptotagmin 4, syntaxin 1, syntaxin 16, synaptobrevin-4/VAMP4, SNAP-25, synaptophysin, Munc 18, intersectin 1, intersectin 2, clathrin, AP-2, AP-180, dynamin 1, dynamin 2, dynamin 3. Whenever specific antibodies were available, we also inspected cellular distribution of aforementioned proteins in endophilin mutants. Although asked to move all these findings to the Supplementary Information, we kept some data in the main Figures since we consider that they are important for the conclusions presented here.

Major comment 6. The authors did not perform experiments the reviewer suggested. Fig. S2K plots the fluorescence level but not the pHluorin spot number, which may not reflect the exocytosis number. The reviewer hope to see that in the presence of pitstop, TKO still reduces the capacitance increase.

Capacitance measurements, and/or amperometry, have been used to demonstrate conclusively the involvement of the three SNAREs, synaptotagmin-1, synaptotagmin-7, Munc18-1, CAPS-1/2, complexin and ubMunc13-2 in chromaffin cell secretion. It is important to appreciate that our data show not only that the capacitance increase is reduced, but also a reduction in amperometric current (measured in parallel), which reports on adrenaline release, in endophilin TKO cells. In addition, pHluorin-based experiments with two independent probes revealed lower secretion from endophilin TKO cells.

The Reviewer suggested to use Pitstop in combination with capacitance measurements. We would like to point out that (1) the existing electrophysiological and amperometry measurements already establish that exocytosis is affected, by the same standards that were previously used to demonstrate the involvement of above-mentioned proteins in exocytosis, and (2) so far, Pitstop has never been used together with capacitance measurements in chromaffin cells since the results might not be easy to interpret. It has been shown in neurons that blocking endocytosis with Pitstop-2 leads directly to reductions in synaptic release, even

before vesicles recycle (Hua et al., Neuron 2013 - PMID: 24139039). The reason is likely a block of the release sites on the plasma membrane (Hua et al., 2013). It is unknown how this will play out in a chromaffin cell, whether there is a time-dependent inhibition of exocytosis, or not. If there is an inhibition it might be different in the case of endophilin TKO cells, since endocytosis kinetics appears to be delayed at early time-points. Thus, it is unclear how to interpret such data: we would need to first investigate the mechanism and timing of Pitstop-2-induced inhibition of exocytosis, which is a large task in itself given that the capacitance measurements on transgenic animals are challenging and time consuming (each experiment takes 3-4 months, depending on the animal availability). Even if additional funding and people for this purpose could be found timely and matings of reproductively-challenged endophilin 1KO-2HT-3KO mice work smoothly, performing these experiments on endophilin TKO, endophilin 1KO-2WT-3KO and WT mice with Pitstop-2 (and control) would delay publication of this work by another 6 months without gaining a clear advantage over the present data. In this particular case, characterizing chromaffin cell capacitance measurements within 5 seconds after stimulation in the presence of Pitstop-2 would bring little clarity.

For convenience, the original revision points addressed between December 2018 and June 2019 are included below

Summary of the main changes included in the revised manuscript:

- We have included WT controls, in addition to the littermate controls, in the electrophysiological and electrochemical experiments in the new Figure 1 and new Suppl. Fig. S1: these data demonstrate that there is no difference between WT and control littermate (endophilin 1KO-2WT-3KO) chromaffin cell secretion, likely due to endophilins' redundancy. Following the norms in this field, for other electrophysiological and electrochemical experiments (new Figures 2, 6, 7, 8, S2 and S6), either the littermate control, or endophilin TKO cells rescued with the re-introduced endophilin expression, are added.
- We have verified the exocytic defect in chromaffin cells without endophilins by an additional independent approach (besides fast capacitance measurements and amperometry recordings). Here, we constructed and produced virus encoding synaptotagmin-1-pHluorin (a LDCV-resident pH-sensitive probe) and established pHluorin-based exocytosis assays in chromaffin cell primary cultures. We found that endophilin TKO cells expressing synaptotagmin-1-pHluorin display a lower fluorescence increase upon stimulation when compared to controls. These data are included in the new Suppl. Fig. S2I-L. Using the pHluorin assay have also examined exocytosis in the presence of the endocytic inhibitor Pitstop-2, as suggested by Reviewer #1. Importantly, these data

show that the difference in pHluorin exocytosis between endophilin TKO and control cells remains in the presence of this potent endocytic inhibitor, indicating that defective exocytosis in TKO cells is unlikely to be an indirect consequence of impaired endocytosis in the absence of endophilins. These data are presented in the new Suppl. Fig. S2K.

- We employed a plasma membrane sheet assay to analyze the distribution of proteins at the cell surface instead of the suggested TIRF microscopy experiments since this assay allows direct access to endogenous rather than overexpressed proteins. Notably, protein overexpression in small mouse chromaffin cells (average cell size \varnothing 6-8 μ m) is feasible only through a viral infection, and overexpression increases the fraction of cytosolic signal and reduces spatial resolution. Thus, to address the points raised by Reviewers #1 and #3 considering endophilin's presence on the LDCVs, we have generated plasma membrane sheets (cell footprints) by un-roofing WT mouse chromaffin cells in culture, and examined endogenous proteins by immunostaining against endophilin-1, or endophilin-2, and CgA (a marker of LDCVs) and intersectin 1. We observed a substantial colocalization of endophilins with both CgA and with endogenous ITSN1 (endogenous signals). As endophilin is partially a cytosolic protein and binds to several other proteins, a complete overlap of signals cannot be expected. These results are presented in the new Figures 1C-D and new Suppl. Fig. S7K (colocalization quantifications are present in Suppl. Fig. S1 and in the text).
- Since quantitative biochemical analysis on isolated LDCVs from endophilin TKO chromaffin cells is not feasible with the available technology (see below), we addressed Reviewer #1's concern about performing the biochemical analysis in adrenal gland homogenate (old Fig5 and Suppl. Fig. S4) as opposed to isolated LDCVs by a more sensitive alternative approach. We isolated RNA from the adrenal medulla and transcribed it into cDNA in order to examine gene expression of several key LDCV integral membrane proteins, LDCV cargo proteins and proteins important for LDCV formation/maturation by real-time quantitative PCR. These data (presented in the new Suppl. Fig. S4E) confirm our complementary immunocytochemistry experiments on chromaffin cells in culture and immunoblots of adrenal gland homogenate, which show that loss of endophilins does not lead to gross alterations in LDCV gene expression.
- Following Reviewer's #1 comment, we have inspected the adaptor protein 2 (AP2) distribution by same approach as used that for intersectins, but were unable to observe significant translocation of AP2 to the plasma membrane. These data are presented below as a Figure for Reviewers. Moreover, we have replaced the original image with a more representative one in the new Figure 5A. Together with our data using Pitstop-2 (Fig. S2K) these experiments suggest that the exocytic defects in LDCV exocytosis in TKO cells are unlikely to be caused by defective endocytic protein function.

- To better understand the mechanism of endophilin's role in exocytosis, we have systematically tested if a complex formation between endophilins and intersectin is required for regulation of chromaffin cell secretion. We have first evaluated the endophilin1:ITSN1 interaction, originally published by Pechstein et al. EMBO Rep. (2015, PMID:25520322), by a pull-down assay (new Figure 8A). Further, based on the ITSN1-SH3B domain structure (Pechstein et al., 2015), we designed an endophilin-binding defective ITSN1 variant (W949E+Y965E) expected to block its interaction with endophilin. Subsequent biochemical interaction studies revealed the lack of interaction between endophilin 1 and the ITSN1 W949E+Y965E mutant, as shown in the new Figure 8A.

To explore the role of the endophilin-intersectin interaction in chromaffin cell exocytosis, we then performed experiments complementary to the data shown in Figure 8 of the original submission (now presented in the new Figure 7G-M). We produced lentivirus expressing ITSN1 WT and ITSN1 W949E+Y965E mutant to attempt a rescue of the exocytic phenotype reported in ITSN1-deficient cells (Malacombe et al., 2006-PMID:16874303; Yu et al., 2008-PMID:18676989). While a rescue of the exocytic phenotype was observed with ITSN1 WT, this was not the case when ITSN1 W949E+Y965E was introduced in the ITSN1 KO cells. These results, in conjunction with the lack of rescue of the exocytic phenotype by full-length endophilin 1 with two point mutations (E329K+S336K) that cannot bind ITSN1, show that the loss of the endophilin1-ITSN1 interaction underlies the exocytic phenotype observed in both mutants. We have presented these data as a new Figure 8B-F, and have altered the text accordingly.

- We have studied the translocation of ITSN1-EGFP from the cytosol to the plasma membrane as suggested by Reviewer #3. We found that ITSN1-EGFP translocation to the plasma membrane is triggered by stimulation (chromaffin cell depolarization in high K⁺), as shown in Movie 5, in agreement with a function of an endophilin-intersectin complex in LDCV exocytosis.
- In addition to the experiments in the adrenal gland, we have performed a detailed characterization of WT, endophilin KOWTKO and endophilin TKO chromaffin cells in culture by electron microscopy (EM). No change in the overall morphology, size and number of LDCVs was observed: these data are comparable to the EM data obtained from chromaffin cells in adrenal glands. Data from cultured chromaffin cells are now shown in the new Figure 3, while chromaffin cells in the glands are now presented in the new Suppl. Fig. S3. Curiously, no change in the number of docked LDCVs or the LDCV distribution near the plasma membrane was observed in chromaffin cells in culture (new Figure 3E-F). Notably, the observed differences in the LDCV distribution near the plasma membrane in adrenal gland are minor, and all physiological experiments were performed on the cultured chromaffin cell model where no difference in the distribution of LDCVs in the proximity of the plasma membrane was observed. Thus, we conclude that the mild alterations in the LDCV distribution, if any, are not the cause of the observed exocytic phenotype.

- We have revised a model (new Suppl Fig S9) to illustrate our proposal that endophilin and intersectin act in tandem to stimulate the priming docked vesicles, likely through intersectin's interaction with SNARE proteins and/or by regulating the local actin network near the plasma membrane. Endophilins 1 and 2 are present on the majority, but not on all neurosecretory vesicles, and the effects of their absence on exocytosis are intermediate (when compared to SNAREs, Munc-13, synaptotagmin-1 and -7), suggesting a regulatory role in exocytosis.

Reviewers' Comments:

Reviewer #1:

Remarks to the Author:

Gowrisankaran et al. propose a novel role of endophilin in exocytosis of LDCVs in this manuscript. Using capacitance recordings combined with amperometric measurements the authors reveal defective LDCV exocytosis, which can be rescued by endophilin A1 and A2 re-expression. The reduced release correlates with a mild redistribution of LDCVs observed in EM analysis. Via immunofluorescence they show insignificant changes of exocytic and endocytic proteins in endophilin triple knockout chromaffin cells that modestly affect endocytic uptake experiments. Lastly, the authors propose that endophilin's role in exocytosis is mediated via SH3-based interaction with intersectin.

The data reported here provide a novel action of endophilin in LDCV release. These results would certainly have the impact for a publication in Nature Neuroscience, as endophilins have been largely implicated as an endocytic protein so far as well as an endosomal/autophagosomal regulator by some of the authors.

We greatly appreciate the Reviewer's #1 careful evaluation of our work and the recognition of its significance. We also thank him/her for a support for the publication in Nature Neuroscience provided the specific questions and concerns are resolved. We agree that a role for endophilin in chromaffin cells exocytosis is a novel, unexpected finding.

However, the data appear to be premature and rather descriptive at this stage and therefore, does not fully support the model that has been proposed. It requires further controls and detailed mechanistic insight how endophilin executes its exocytic function to convince the scientific audience.

Generally, the manuscript is clearly written and structured. Specific points, suggestions and questions are summarized below.

We have added a number of additional new experiments; most notably, we confirm the key finding that loss of endophilin directly impairs LDCV exocytosis by pHluorin assays and have added further experiments aimed at addressing the molecular mechanisms underlying the observed defects. We hope that the wealth of new data will convince the Reviewer that our study is not premature and will withstand the test of time. Please see below our detailed response to the specific comments and concerns.

Major comments:

1. The authors should consistently compare the TKO data to the same controls (WT and KO-WT-KO) throughout the manuscript. Especially as they claim that the endocytic uptake defect is only mild and not significant compared to WT (Fig 6), they need to show WT controls in their exocytic measurements as well. The same applies to all rescue experiments: The controls need to be included for statistical comparison in Fig 3, 7, 8 and S3 in all plots.

We agree with Reviewer #1 about the need for additional controls. We have striven to include WT controls (in addition to littermate controls) whenever feasible, e.g. electron microscopy, immunocytochemistry, Western blots, endocytic assays, etc. Electrophysiological and electrochemical experiments were made by adhering to the following norm in the field: Equal numbers of chromaffin cells were recorded from endophilin KOWTKO and TKO (littermates) mice each day. Since we need to mate endophilin 1KO-2HT-3KO mice with each other to obtain reasonable numbers of TKO mice, it is impossible to obtain WT animals as littermates. Following the Reviewer's advice, we took the next best strategy and have now included as WT controls results from WT cells obtained from a separate mouse line (C57BL6/J; same line background as endophilins) that were recorded at the same time as endophilin mutants and on the same setup. Following the conventions in the exocytosis field, we abstained from making direct statistical comparisons between data sets that were not derived from littermates. Driven by Reviewer's suggestion, the 'WT-level' of the different exocytic parameters has now been included in Figure 1 and Suppl. Fig. S1 and were found to be consistent with the endophilin KO-WT-KO controls. We have noted in the Methods and Figure legends that the WT data are collected from independent litters and culture preparations. For other electrophysiological and electrochemical experiments (new Figures 2, 6, 7, 8, S2 and S6), either the littermate control or endophilin TKO rescued with the re-introduced endophilin expression, are added.

2. The release phenotype in TKO chromaffin cells is the major claim of this manuscript. The authors describe it with elegant capacitance and amperometric measurements. However, showing the release defect of LDCVs with an alternative method e.g. pHluorin based recordings, would greatly add to the paper and would strengthen their assertion of a role in exocytosis. Furthermore, can the authors comment on how their exocytic defects compare in magnitude to known exocytic mutants?

Of note, the combination of capacitance and amperometry measurements is one of the most advanced and comprehensive strategies to analyze chromaffin cell exocytosis with the necessary high temporal resolution. Fast capacitance measurements are generally preferable to pHluorin measurements for showing a general exocytotic defect, because they determine the exocytotic capacity of the entire cell, whereas pHluorin assays are typically restricted to one optical plane (when performed with a confocal microscope), or the footprint (when performed in TIRF microscope). Even though Reviewer #2 and #3 were convinced by our combination of electrophysiological and electrochemical data, we agree with Reviewer #1 that obtaining independent verification with another method is always an advantage. Given that we are familiar with the suggested pHluorin experiments (e.g., Milosevic et al, Neuron 2011), we constructed and produced lentivirus encoding

synaptotagmin-1-pHluorin and established the pHluorin-based assay in primary culture of chromaffin cells (viable only for 3-4 days). We first verified this assay under two stimulation conditions (only data with high K⁺-triggered exocytosis are shown here) and in presence of the endocytosis inhibitor Pitstop-2 (new Suppl. Fig. S2I-K). We found that endophilin TKO cells expressing synaptotagmin-1-pHluorin show a lower fluorescence increase upon stimulation than WT and littermate controls (new Suppl. Fig. S2L), thereby confirming that loss of endophilins in chromaffin cells impairs exocytosis.

Others and we have measured exocytosis in chromaffin cells of numerous exocytic mutants, or upon various treatments that affect exocytosis (e.g. PI(4,5)P₂ level modulation, phosphorylation inhibition, etc.): e.g. Sørensen et al., Cell 2003 - PMID:12859899; Milosevic et al., J Neurosci 2005 - PMID:15758165; Nagy, Milosevic et al., MBC 2005 - PMID:16195346; Sørensen et al. EMBO J 2006 - PMID:16498411; Gulyás-Kovács et al., J Neurosci 2007 - PMID:17687045; Liu et al., 2008 - PMID:18495893; Schonn et al., 2008 - PMID:18308932; de Wit et al Cell 2009 - PMID:19716167; Man et al., 2015 - PMID:26575293; Borisovska et al., 2005 - PMID:15920476. While the exocytic defects observed in endophilin TKO cells are substantial, they are not as striking as the loss of the main SNAREs or Munc18-1 (the key components of the exocytic machinery), or the simultaneous loss of both major calcium-sensors synaptotagmin-1 and -7. These data indicate that endophilins are potent regulators of secretion but not essential for exocytosis, as discussed in the revised manuscript.

3. The authors claim in their discussion that endophilin can be found on SVs and LDCVs and is acting during exocytosis. Yet, the corresponding figure showing endophilin on purified LDCVs is missing in the manuscript. To support their proposed mechanism the authors should visualize the presence of endophilins during exocytosis. Recording the localisation of fluorescently tagged endophilin and/or intersectin during release events in time-lapse experiments using TIRF microscopy would strongly support their model.

We agree with the Reviewer, and we had originally attempted TIRF-based experiments in endophilin TKO cells in 2012-2013, and again 2016, yet there was a combination of technical issues that prevented further advances along this line. Mouse chromaffin cells are round primary cells whose cell footprints are very small (often ~3 μm, or smaller), so it was difficult to reliably follow the few exocytic events by a TIRF microscope. The bigger issue, however, are (i) the nature of endophilin, which is a partially cytosolic protein, and (ii) the difficulty to achieve efficient transfection of chromaffin cells, which can only be obtained by virus. When expressed by lentivirus (we have also tried Semliki Forest Virus), the cytosolic pool of endophilin was abundant and has masked endophilin dynamics in the proximity of the plasma membrane.

We have thus optimized an alternative approach by generating plasma membrane sheets from WT chromaffin cells, as in Milosevic et al., 2005 - PMID:15758165 and Nagy, Milosevic et al., MBC 2005 – PMID: 16195346. This assay capitalizes on the direct access to the thin, flat inside-out plasma membrane sheets that still contain elements of cell cytoskeleton and attached vesicles. From studies that also include electron microscopy data, it is known that only structures and proteins that directly interact with the plasma membrane lipids or

proteins (e.g. docked/primed LDCVs) remain attached to the plasma membrane sheets, while cytosolic proteins and organelles are lost (e.g., Avery et al, JCB 2000 - PMID:10648564; Lang et al, EMBO J 200 1- PMID:11331586). We generated, fixed and immunostained such plasma membrane sheets with anti-CgA (marker of LDCVs) and anti-endophilin 1 or anti-endophilin 2 specific antibodies, respectively. The plasma membrane was visualized in the presence of blue (TMA-DPH) membrane dye. We found significant colocalization between endogenous CgA-positive LDCVs and both endogenous endophilin 1 and 2, demonstrating that endophilin 1 and 2 are present on docked/primed LDCVs that will, upon stimulation, fuse with the plasma membrane. These results are presented in the new Figure 1C-D.

It may be of interest to mention here that we have assayed endophilin 1 and 2 localization on LDCVs purified from bovine adrenal glands (a schematic of LDCV purification is shown in panel A, Figure on the right). We found that endophilin 1 indeed is present on LDCVs by Western blot (panel B – Figure on the right; note that equal protein concentrations were loaded into each lane: endophilin 1 is a brain-enriched protein so its signal in adrenal gland homogenate is barely visible under these conditions). Due to a technical issue with a new lot of anti-endophilin 2 antibody, we were not able to test for endophilin-2 in these experiments during 2017 when bovine material was readily available; and due to new regulations we are no longer able to obtain bovine adrenal glands from local slaughterhouses. We did not included these experiments in the manuscript since the data for endophilin 2 are missing. Purification of LDCVs from mouse adrenal glands is not feasible due to the gland's small size in mice (see below). Notably, we also consider that plasma membrane sheet experiments are more conclusive since endophilin's presence could be examined at the docked/primed LDCVs.

4. For Fig 5 and FigS4, Gowrisankaran et al. describe the LDCV protein composition using immunofluorescent stainings in order to exclude an exocytic defect due to altered LDCV formation and altered exocytosis protein expression. This is a good start, but not sufficient and requires a quantitative biochemical analysis of these proteins in WT/TKO chromaffin cells (as opposed to adrenal gland homogenates) and also western blot analysis from purified LDCVs where relevant (for .e.g For CgA and Syt-1).

It would be exceptionally useful to do the experiments as suggested by the Reviewer, but these experiments are not feasible at present due to technical limitations. Endophilin litters are small (6-8 pups) and sparse since endophilin KO-HT-KO animals (parents of endophilin

TKO pups) have epilepsy, neurological issues and mate poorly (Milosevic et al., Neuron 2011; Murdoch et al. Cell Rep 2016). Notably, endophilin TKO mice are born in 1:4 ratio and die at birth. At best, we get 1-2 TKO animal(s) per litter that need(s) to be prepped fast, without knowing the genotype. The diameter of the adrenal glands from a newborn mouse is smaller than ~1 mm, and only a few thousand chromaffin cells are located within the gland's medulla (the other main cell type in the adrenal gland are cortical cells). While we have managed to collect enough cells from the adrenal medulla of newborn pups to perform real-time qPCR (after pooling cells from 2-4 glands from 1-2 TKO mice after genotyping - see Suppl. Fig. S1A and new Suppl. Fig. S4E), there was not enough material to perform Western blot or proteomics from endophilin TKO chromaffin cells (we have successfully performed Western blots only when using the whole adrenal gland, as in Suppl. Fig. S1B, Figure 5B, Suppl. Fig. S4A-D, Suppl. Fig. S7A-B).

Considering the technical challenges, the small size of newborn mouse adrenal glands being the biggest of all, it is impossible to isolate LDCVs from this material by any established protocol. We, and others in the field, have purified LDCVs from 3-4 adrenal glands isolated from two adult cows (size of each gland is ~4x5cm) in the past. We estimate that one would need ~13,500 endophilin TKO pups to reach the starting material for one experiment (optimization will be needed). Since endophilin TKO mice come 1:4, we would need to generate and genotype ~54,000 mice, and it is not clear if the frozen material would be of much use in this case. The quantitative biochemical analysis of LDCVs from endophilin TKO pups is thus not possible until further technology is developed.

However, we collected enough material for additional qPCR experiments, and tested gene expression of several key proteins important for LDCV formation/maturation, trafficking and exocytosis by this method. The results, presented in the new Suppl. Fig. S4E, show no difference in the mRNA levels of the key LDCV integral proteins, LDCV cargo proteins or the proteins known to be important for LDCV formation/maturation.

5. The authors state on page 9: "... the distribution of AP2 and dynamin seemed unaltered in endophilin TKO cells...". This statement is unclear. The authors should perform a thorough analysis as shown for intersectin localisation to come to a clear conclusion. Is the localisation of AP2 redistributed to the plasma membrane? The representative images suggest a change in localisation.

We have looked more carefully into AP2 distribution by the same quantification approach as used for ITSN1/2, but we did not observe significant translocation of AP2 at the plasma membrane under basic conditions - the outcome of such an analysis is shown in the Figure on the right. This is now reported in the text, and a more representative image is shown in the new Figure 5A.

6. To dissect the roles of endophilin in exocytosis and endocytosis the authors should block endocytosis e.g. by using pitstop and measure exocytic release in controls and TKO chromaffin cells. Thereby they can exclude any contribution from endocytic processes. Moreover, the authors should elucidate how endophilins actions during exo- or endocytosis are regulated. Are there two different endophilin pools or the same endophilin molecules are required for membrane retrieval and release?

We agree with the Reviewer about the need for additional justification of the statement that endophilins' role in exocytosis is independent from their role in endocytosis. Using synaptotagmin-1-pHluorin-based exo-endocytosis assays in chromaffin cells (see point #3), we have examined the fusion of LDCVs in the presence of the endocytic inhibitor Pitstop-2 in WT, endophilin KOWTKO and endophilin TKO chromaffin cells. These new data demonstrate that the difference in pHluorin exocytosis between TKO and control cells remains in the presence of this potent endocytic inhibitor, indicating that defective exocytosis in endophilin TKO cells is unlikely to be an indirect consequence of impaired endocytosis in the absence of endophilins. These data are presented in the new Suppl. Fig. S2K and explained in Results. Nonetheless, we note that it is difficult to completely uncouple exocytosis from endocytosis in any living model system (e.g. due to limited site clearance).

To our knowledge, different endophilin pools were not reported before. Bai et al. Cell 2010 - PMID:21029864 have suggested that endophilin is delivered for endocytosis by exocytosis in *C. elegans*, a much simpler model that has only one endophilin-A protein. While we consider that a pool of endophilin present on the secretory vesicle could be used for endocytosis after the previous round of exocytosis, it would be very difficult to directly demonstrate that in the chromaffin cell model system. We have now expanded our Discussion to elaborate this point.

7. The authors present an SH3-mediated interaction of endophilin with intersectin as a model to explain the observed phenotypes. This model requires further experimental evidence before publication.

To resolve the conceptual issue raised by the Reviewer and to emphasize one of the three main findings of this study, we have performed several additional experiments and adjusted this part of the manuscript through extensive revisions of the text.

a) A rescue experiment using the endophilin SH3 domain fused to a non-endophilin BAR domain would emphasise the importance of the SH3 domains in the exocytic process, while the rescue experiment with the BAR domain only is not sufficient in Fig 7. Can you explain the dominant-negative effect of the Bar-domain only rescue in the context of the model?

We acknowledge the Reviewer's remark that the rescue experiment with the endophilin BAR domain is not sufficient. We did not attempt the rescue with endophilin's SH3 domain only since these experiments would not be physiological given that the expression of the SH3 domain alone alters many cellular interactions (for example, see Grab et al 1997 - PMID:9148966; Shupliakov et al, 1997 - PMID:9092476). We considered swapping endophilin's BAR domain for other N-BAR domains, yet endophilin's BAR domain also contains an amphipathic helix and has special curvature-sensing properties, and we assume

that it is uniquely suited for the protein's role in exocytosis. We also have to consider that swapping the BAR-domains and testing for rescue in the somewhat complex setting of our model system (chromaffin cells from newborn mice, lethal triple knockout mice, viral transduction and single-cell capacitance and amperometry measurements, etc.) would require a large number of mice, while the outcome would not allow any clear interpretations, since it is a priori unknown whether any other BAR domain can substitute for that of endophilin.

We thus took an alternative approach to address this conceptual issue. It has been published that, similar to chromaffin cells lacking endophilin, cells without intersectin-1 also show reduced exocytosis (ITSN1 KD: Malacombe et al., 2006 - PMID:16874303; ITSN1 KO: Yu et al., 2008 - PMID:18676989). Furthermore, the magnitude of exocytic defects in ITSN-1 KO chromaffin cells (Yu et al., 2008) is comparable in magnitude to the endophilin TKO mutant (our study). We have published earlier that ITSN1 interacts with endophilin through its second SH3 domain (for the characterization of this interaction and the NMR structure, see Pechstein et al., 2015 - PMID:25520322). Based on the NMR structure data, we have designed a mutant ITSN1 protein carrying two point mutations (W949E+Y965E) that interfere with endophilin 1 binding. Our new biochemical data reveal that ITSN1-W949E+Y965E indeed fails to bind endophilin1 (see new Figure 8A). Importantly, the expression levels of ITSN1 WT-EGFP and ITSN1 W949E+Y965E-EGFP were comparable (Suppl. Fig. S7J) – suggesting that the expression and stability of the mutant protein is similar to the WT protein. ITSN-1 WT and ITSN-1 W949E+Y965E were then expressed in ITSN-1 KO cells using lentiviral vectors and exocytosis was assayed. These new data shown in the new figure 8 show mutant endophilin binding-defective ITSN-1 W949E+Y965E fails to rescue defective exocytosis in ITSN-1 KO cells.

These results, taken together with a lack of rescue by ITSN1-binding defective endophilin, establish that endophilin-ITSN complex formation is of key importance for LDCV exocytosis. We have presented these data as an entirely new Figure 8, and have altered the text accordingly.

Considering the dominant-negative effect of the BAR-domain only rescue, we acknowledge that the overexpression of endophilin's BAR-domain to attempt a rescue is not an ideal experiment (see above). The dominant-negative effect of the BAR-domain overexpression likely arises from a competition with other proteins peripherally attached to vesicles and/or interference with endophilin interactors. This aspect is now included in the manuscript.

b) If intersectin is recruited to LDCVs via endophilin, the authors should show purified LDCVs derived from TKO mice devoid of intersectin.

We agree that this would be a very instructive experiment, however, it is not technically feasible, as detailed in the point #4.

c) Their model suggests that intersectin mislocalisation is blocking release of LDCVs in absence of endophilin. To validate this hypothesis the authors should show whether the intersectin clusters at the plasma membrane colocalize with LDCV release sites.

This is not an easy question to answer in this model system since chromaffin cells in culture lack dedicated release sites. Instead, LDCVs appear to fuse all over the surface of the

chromaffin cell, effectively generating their own fusion sites as they fuse. In addition, experiments with endogenous proteins are not feasible in this case since the specific antibodies against CgA and intersectin are raised in the same animal species (rabbit), and virus-driven intersectin expression (the exclusive way of chromaffin cell transfection) required for live cell imaging is not ideal to study intersectin's localization in the plasma membrane. We thank Reviewer for this suggestion, which we considered originally as well, yet is not trivial to link intersectin's clusters at the plasma membrane to LDCVs fusion sites.

d) Could the authors please comment on why the EndoA1ΔITSN mutant can only mimic the defective capacitance burst release shown in TKOs, but not the sustained release? (Fig 8)

We have performed additional experiments here (expression of both endophilin 1 WT and endophilin 1 E329K+S336K - see new Figure 7G-M), and the small differences in the sustained component observed originally are no longer significant.

That said, the standard interpretation of the sustained component is that it reports on ongoing priming as the Ca²⁺-concentration is elevated following stimulation (such that fusion triggering is not limiting). In contrast, the RRP and SRP report on the standing pool sizes, which also depend on the reverse priming reaction (unpriming). Endophilin appears to both increase forward priming, but also decrease unpriming, thereby building up larger pools. Yet, only the latter reaction depends on its binding to intersectin. This is born out in intersectin-based experiments, where WT and mutant ITSN1 rescue cells do not differ in their sustained component, but they differ in pool sizes. So, a separate ITSN1-independent function of endophilin may be to mediate the increase in forward priming. We have now inserted a brief note on this in the manuscript.

e) Given intersectin's interaction with actin assembly proteins such as Cdc42 and N-Wasp and its redistribution to the plasma membrane under steady state conditions, can the authors explain why actin density is reduced in resting conditions?

Actin has a complex role in the organizing/controlling the secretory machinery in neuroendocrine cells. ITSN1 indeed interacts with actin assembly proteins Cdc42 and N-WASP, and has a GEF domain via which it can activate Cdc42 to trigger actin assembly at/near the plasma membrane (Hussain et al., 2001-PMID:11584276). However, in line with the important function of F-actin in numerous cellular processes, actin regulatory proteins are subject to intricate layers of regulation. Consistently, ITSN1's GEF activity is not constitutively switched on, but rather auto-inhibited by the short SH3-DH domain linker (Ahmad & Lim, 2010 - PMID: 20585582). Therefore, an accumulation of ITSN1 near plasma membrane is not expected to automatically lead to an increase in actin's density since the auto-inhibition would first have to be relieved by the appropriate regulatory input. Maybe endophilins, or endophilins' interactors, are required in this step. In addition, mislocalized scaffold protein ITSN1 may also mislocalize/sequester away components needed for actin regulation. However, at this point these are only speculations and a follow-up study is needed in the future. Thus, only a possibility that actin plays a role here is raised in this manuscript.

f) The authors find that Intersectin is more at the membrane in the TKO vs WT. but wouldn't this also affect endocytosis? In the discussion (Page 13) they themselves state that intersectin couples endo- and exocytosis.

This is an open question. Notably, we do not detect major endocytic defects in endophilin TKO chromaffin cells, however, mislocalized intersectin might not be at the correct site for endocytosis. We have now mentioned this point in the Discussion.

g) The authors should perform EM studies after the expression of EndoA WT and Δ ITSN to show that WT rescue can restore LDCV distribution near the membrane.

To be able to perform virus-based rescues and to make these experiments comparable to the electrophysiological and electrochemical experiments done on cultured chromaffin cells, we have first completed a detailed ultrastructural characterization of resting WT, endophilin KO-WT-KO and endophilin TKO chromaffin cells in culture (DIV 3-4). In this completely independent dataset, we observed no change in the size of LDCVs and their overall numbers (comparable to our EM data obtained in the adrenal glands). However, we also did not detect a significant change in LDCV distribution near the plasma membrane. It is well accepted in the field that the distribution of LDCVs may be altered between chromaffin cells in the adrenal gland and cultured chromaffin cells – in line with this, cultured chromaffin cells, for example, display a rounder shape and seem to lose the hot-spots of exocytosis. In addition, the differences in the LDCVs distribution near the plasma membrane in adrenal gland chromaffin cells were minor, and all physiological experiments were performed on the cultured chromaffin cell model where no difference in the distribution of LDCVs in the proximity of the plasma membrane was observed. Thus, we concluded that the difference in distribution of LDCVs in the proximity of the plasma membrane is not relevant for the observed exocytic phenotype. This conclusion is now presented in the revised manuscript.

Minor comments:

In the material and methods, a section describing the western blot procedure and analysis is missing. Please include it.

Thank you for noticing that the Western blot procedure and analysis parts were missing. We have now included them in the Method section. Due to this mishap, it was probably difficult to realize the complexity of these experiments and the difficulties that arise simply from the small size of adrenal glands from a newborn transgenic mouse.

Every figure should be understandable on its own. Thus, please label all bar graphs with the corresponding genotypes.

We hope that our updated labeling is satisfactory.

Data from the KWK genotype is presented in Fig 1B but the description of what KWK means is only described in the results section for Figure 1. Please move it to an earlier part of the manuscript.

The description of KWK is included earlier in the manuscript.

For all line profiles please show the ROI used for quantification in the corresponding image.

We have now included the ROI used for quantification in the corresponding images for all line profiles – thank you for noting this point.

Pg 6 – why is the lentiviral expression verified in HEK cells (Fig S3A)

The combination of electrophysiological and electrochemical experiments is time-demanding: after (genotype-blind) preparation of primary chromaffin cells in culture and subsequent genotyping (1-2 days) we can, at best, record just 1-3 cells per hour and only several cells per condition (note that the same number of cells from each condition is recorded on the same day, and that we usually test 3-4 conditions). Thus, it is instrumental to verify the lentivirus quality and check for proper protein expression, protein size and stability. It would be ideal to verify the expression in cultured chromaffin cells, and we do that by quantification of fluorescence images of cells with the EGFP expression (through a bicistronic viral system). Yet, the yield/material output from cultured chromaffin cells expressing lentivirus is not sufficient to perform the Western blotting reliably, so in addition, we use HEK293 cells that express our protein of interest for a time-period similar to that used in chromaffin cells. We typically use clonal cultured cells (e.g. HEK293 or PC12 cells) since we did not notice a significant difference in the efficiency of transfection and protein expression between HEK293, PC12 or chromaffin cells. Together, these two experimental sets provide us with the necessary confidence in the used lentiviral preparations before we initiate the complex and time-consuming electrophysiological and electrochemical experiments.

The figure legend of S6 does not fit to the figures. Moreover, please explain abbreviations in Fig S6 G more clearly.

We have altered Suppl. Figure S6G (new Suppl. Figure S7G-H) in such a way that a schematic of LDCV isolation is now included (Suppl. Figure S7G), and the figure legend now contains explanations of all abbreviations. We have also extended the Method section to detail the procedures of SVs and LDCVs purifications.

Page 4 line 116: “ ... with the LDCV maker, ...” should be “ ... with the LDCV marker, ...”

Done.

Page 14 line 454: “... (Supple Figure S7) Thus, lack...” should be “... (Supple Figure S7). Thus, lack...”

Done.

Page 15 line 498: “... our study suggests that endocytosis...” should be “... our study suggests that exocytosis...”

Done.

Reviewer #2:

Remarks to the Author:

This paper investigates the role endophilin-type proteins for release of large dense core vesicles in chromaffin cells. Using a combination of amperometry, capacitance measurements and exploiting the previously established triple knock outs for all three mouse endophilin loci, they demonstrate a role for Endophilins in vesicle EXOCytosis and proper size of RRP and SRP pools. Mechanistically, they show that the single SH3 domain of Endophilins is important for rescue activity, and that a double point mutation previously shown to interfere with the SH3-SH3 interaction between Endophilin and Intersectins diminishes rescue activity. They explore the integrity of release machinery, however, without identifying mechanistic links to decreased exocytosis. The authors describe an interesting observation, which is that Endophilins might execute a exocytotic role for neurosecretory vesicles besides the well established endocytotic role these proteins play in the synaptic vesicle cycle. Their work concerning the molecular-genetic analysis and also particularly the electrophysiological analysis is convincing.

We appreciate that Reviewer #2 finds our work interesting, and the “molecular-genetic analysis and also particularly the electrophysiological analysis convincing”.

However, it remains to be seen that no mechanistic explanation of the phenotype they describe can be provided. Even if Intersectin was involved, how do we come from here to the actual exocytotic process? I am afraid that without at least the beginning of such a mechanistic explanation, the manuscript falls short of being of general interest enough to justify publication in Nature Neuroscience.

The reviewer raises the important point regarding the molecular mechanism by which endophilins regulate exocytosis. In our revised manuscript we have addressed this issue in several different ways:

(i) Analysis of other pathways indirectly linked to LDCV exocytosis: (1) We have isolated RNA from the adrenal medulla and transformed it to cDNA in order to examine gene expression of several key LDCV integral membrane proteins, LDCV cargo proteins and proteins important for LDCV formation/maturation by real-time quantitative PCR. These data (presented in the new Suppl. Fig. S4E) confirm our complementary immunocytochemistry experiments on chromaffin cells in culture and immunoblots of adrenal gland homogenate, which show that loss of endophilins does not lead to gross alterations in LDCV gene expression. (2) We performed a detailed characterization of WT, endophilin KOWTKO and endophilin TKO chromaffin cells in culture by electron microscopy (EM). No change in the overall morphology, size and number of LDCVs was observed: these data are comparable to the EM data obtained from chromaffin cells in adrenal glands. (3) We have examined the fusion of LDCVs in the presence of the endocytic inhibitor Pitstop-2 in WT, endophilin KOWTKO and TKO chromaffin cells. These new data demonstrate that the difference in pHluorin exocytosis between TKO and control cells remains in the presence of this potent endocytic inhibitor, indicating that defective exocytosis in endophilin TKO cells is unlikely to be an indirect consequence of impaired endocytosis in the absence of endophilins. These data are presented in the new Suppl. Fig. S2K and explained in Results.

(ii) Analysis of endophilin localization: We found significant colocalization between endogenous CgA-positive LDCVs and both endogenous endophilin 1 and 2, demonstrating that endophilin 1 and 2 are present on many docked/primed LDCVs that will, upon stimulation, fuse with the plasma membrane. The results are presented in new Figure 1C-D and show that endophilins are required locally on the LDCV itself to mediate fusion, e.g. by its membrane deforming activity or, indirectly, by association with other proteins, most notably with intersectin (see below).

(iii) Endophilin association with other proteins known to have a role in exocytosis, e.g. intersectin. A careful examination of known endophilin-binding partners revealed a close physical and functional association of endophilin with intersectin, a protein known to have a function in coupling exo- and endocytosis at synapses (Sakaba et al., 2013-PMID: 23633571) and, of particular importance for this study, a direct role in the chromaffin cell exocytosis (e.g. Malacombe et al., 2006 - PMID:16874303, Yu et al., 2008 – PMID: 18676989). In the revised manuscript, we have corroborated these findings in several important ways. Namely, the magnitude of exocytic defects in ITSN1 KO chromaffin cells (Yu et al., 2008) is comparable in magnitude to the endophilin TKO mutant (our study). We have published earlier that ITSN1 interacts with endophilin through its second SH3 domain (for the characterization of this interaction and the NMR structure, see Pechstein et al., 2015 - PMID:25520322). Based on the NMR structure data, we have designed a mutant ITSN1 protein carrying two point mutations (W949E+Y965E) that interfere with endophilin 1 binding. Our new biochemical data reveal that ITSN1-W949E+Y965E indeed fails to bind endophilin1 (see new Figure 8A). Importantly, the expression levels of ITSN1 WT-EGFP and ITSN1 W949E+Y965E-EGFP were comparable (Suppl. Fig. S7J) – indicating that the expression and stability of the mutant protein is similar to the WT protein. ITSN1 WT and ITSN1 W949E+Y965E were then expressed in ITSN-1 KO cells using lentiviral vectors and exocytosis was assayed. These new data (shown in new Figure 8) show mutant endophilin binding-defective ITSN1 W949E+Y965E fails to rescue defective exocytosis in ITSN1 KO cells. This complementary experiment, taken together with a lack of rescue by ITSN1-binding defective endophilin (E329K+S336K; new Figure 7G-M), establishes that endophilin-ITSN complex formation is of key importance for LDCV exocytosis. We have presented these data as an entirely new Figure 8, and have altered the text accordingly.

Intersectin, conceivably based on prior data and its rich interactome, could regulate LDCV exocytosis by several non-exclusive activities including (1) direct association with the SNARE machinery based on its reported binding to SNAP-25 or SNAP-23 (Okamoto et al., 1999-PMID:10373452), (2) local nanoscale regulation of the F-actin based cytoskeleton via Cdc42 activation by its DH-PH domain, and/ or (3) local membrane deformation by regulating endophilin's BAR domain activity. We discuss these possibilities in the revised manuscript, while years of future studies will be required to examine all of them in detail. Kindly note that with respect to other important regulators of exocytosis such as Munc13 or complexins it took many years, often decades of study with series of papers published in high impact factor journals to unravel their mechanism of action (and for some of them even now no consensus has been reached). We therefore argue that the rich dataset we present here serves as a door opener for future studies to test the above possibilities in more detail. We

hope that Reviewer will agree that our presented findings are of interest to the general neuroscience community and qualify for publication in Nat Neurosci.

Reviewer #3:

Remarks to the Author:

This study reported a novel role of Endophilin-A in neurosecretory vesicle recruitment, priming and fusion besides its classic function in endocytosis. TKO of endophilin resulted in reduced exocytosis, smaller vesicle pools and changed fusion kinetics. These defects can be rescued by knock-in of endophilin A1 and A2. Further analysis indicates SH3-domain and, partially interaction with intersectin, are the underlying mechanism. The presented data are interesting and show several important findings. However, some methodological drawbacks explained below should be resolved before publication.

We thank the Reviewer for finding our work interesting, important and worth publishing. The methodological drawbacks that were pointed out have been addressed in the revised version, as detailed below.

Major concerns:

1. The capacitance jumps (total and RRP capacitance) in the same TKO condition are significantly different in different figures. For example, for total Cm Jump, Fig. 2B red bar is ~350 fF, Fig. 3B red bar is ~220 fF, Fig. 7B red bar is ~380 fF. These differences are similar to the differences between wild-type and TKO shown in Fig. 2B. If the same control data are significantly different among different groups, it is difficult to evaluate the effect of TKO. Similarly, For RRP Cm jump, Fig. 2E TKO red bar is ~75 fF, 3E red bar is ~30 fF, whereas 3E TKO-endoA1 green bar is ~85 fF. Comparing Fig. 3E between TKO (~30 fF) and TKO-endoA1 rescue (~75 fF), TKO-endoA1 rescue indeed rescued the RRP jump. However, if Fig. 3E TKO-endoA1 is compared with Fig. 2 TKO (~75 fF), the rescue is minimal. The variation among different data sets makes it difficult to evaluate the effect of the rescue. This issue needs to be clarified throughout the manuscript.

The capacitance measurements were made over a span of 6 years. Under these circumstances, variability from dataset to dataset is expected, e.g. due to alterations in the electrophysiology setups or experiments performed by different scientists. This is a general finding in physiology (although sometimes it remains ‘hidden’ from the reader, due to normalization or – in some cases – data selection). To circumvent this problem we adopt the strict standard of always doing experiments (for instance WT and mutant rescue) in a parallel fashion: the two conditions are always compared side-by-side, while constructs are expressed in cells from the same animal, and the experimenter tries to obtain the same number of cells for each condition on each experimental day. When comparing two genotypes (KO and WT), this is done exclusively in littermates, prepared in parallel and recorded on the same day. Therefore, we are confident that conditions measured in parallel can be compared (these would be the conditions presented in the same figures), but there is no guarantee that comparisons can be made between experimental series (this would also violate the ‘random sampling’ assumption of standard statistical tests), although sometimes

this is seen in the literature. We have now described this explicitly in the text (Methods, and briefly in the Results and Figure legends).

2. In Fig. 4, instead of fixing the whole adrenal gland, the EM data should be collected from cultured cells where electrophysiological data were collected and compared with the EM data. Otherwise, electrophysiological data should be collected from the whole adrenal gland to be comparable.

We gladly followed the Reviewer's suggestion: We have performed EM studies on chromaffin cells in culture in order to directly compare ultrastructural data to the electrophysiological and electrochemical experiments. The results of the systematic EM characterization of cultured WT, endophilin KOWTKO and endophilin TKO chromaffin cells are shown in the new Figure 3 (data from chromaffin cells in glands are shown as new Suppl. Fig. S3). We observed no change in the morphology and size of LDCVs as well as in their overall numbers (comparable to our EM data obtained from the adrenal glands). Yet, we did not observe any change in the LDCV distribution near the plasma membrane when chromaffin cells from endophilin TKO and its control (WT from C57BL6/J line and littermate KOWTKO) newborn mice were cultured. It is well accepted that the distribution of LDCVs may be altered between chromaffin cells in adrenal gland and cultured chromaffin cells – for example, cultured chromaffin cells display a rounder shape and seem to lose the hot-spots of exocytosis. Given that the differences in the LDCVs distribution near the plasma membrane in adrenal gland were minor, and all physiological experiments were performed on the cultured chromaffin cell model where no difference in the distribution of LDCVs in the proximity of plasma membrane was observed, we conclude that the difference in distribution of LDCVs in the proximity of the plasma membrane, if any, is not relevant for the observed exocytic phenotype.

3. The part on endophilin regulation on intersectin is weak. Following points need to be addressed: It's important to know whether ITSN-1/2 translocation from cytosol to the plasma membrane occurs before or after depolarization, therefore, ITSN-1/2 kinetics need to be presented. A colocalization test needs to be done on endogenous ITSN-1/2, endophilin and CgA in resting condition to show the interaction/recruitment of ITSN-1/2 by endophilin on the vesicle. In addition, co-immunoprecipitation test needs to be done to confirm the interaction between ITSN-1/2 and endophilin. Concerning ITSN-1 function, although previous research (Yu et al., 2009) has already shown that ITSN-1 regulates chromaffin cell exocytosis, at least a ITSN-1 knockdown experiment needs to be done to show that ITSN-1 indeed regulate exocytosis in term of capacitance recording and amperometry. In Figure 3B-G, 3J-Q, and 8H-N, please add another control bar (KWK) to show how much the EndoA1/EndoA1-ΔITSN can/cannot rescue the phenotype. In addition, in Figure 8H-N, besides the TKO bar and TKO+EndoA1-ΔITSN bar, please add the control bar (KWK) and TKO+EndoA1 bar for comparison.

We thank the Reviewer for these excellent suggestions. We have performed several new experiments, which have revealed the following:

- 1) The translocation of ITSN1-EGFP from the cytosol to plasma membrane only occurs during chromaffin cell stimulation, as the new Movie 5 shows. Yet, our temporal resolution is limited to 300 ms (imaging rate) in these experiments.
- 2) We have adapted a plasma membrane sheet assay to mouse chromaffin cells in order to study the colocalization between endogenous endophilins 1 and 2 and CgA, as well as between endophilins 1 and 2 and ITSN1 in the resting condition (see also Reviewer #1's point 3). Due to a lack of specific antibodies raised in different species, the triple colocalization of endogenous CgA-endophilins1/2-ITSN1 proteins, or the colocalization between CgA and ITSN1/2 proteins, was not feasible. The data are presented in the new Figures 1C-D and new Suppl. Fig. S7K, and the text is updated accordingly to report on a significant colocalization detected between endophilins 1 and 2 and CgA, as well as between endophilins 1 and 2 and ITSN1, in the resting condition. Note that both endophilins and intersectins have multiple interactors and functions at the plasma membrane, thus, a complete overlap of signals cannot be expected.

In addition, we show that both ITSN1 and ITSN2 are present on purified LDCVs, as well as on purified SVs. These data are presented in the Suppl. Fig. S7G-H.

- 3) Endophilin 1 was shown to interact with ITSN1 originally in Pechstein et al., EMBO Rep (2015). We have performed additional biochemical assays in order to verify this interaction, including the complementary experiments. In short, we have identified two-point mutations (W949E and Y965E) in the intersectin-1 SH3B domain, and found that they abolish ITSN1's interaction with endophilin 1 (shown in the new Figure 8A).
- 4) Instead of the proposed ITSN1 knock-down experiment, we have tested exocytosis in ITSN1 KO chromaffin cells provided by V. Haucke's laboratory. Similar to endophilin TKO cells, ITSN1 KO chromaffin cells show reduced secretion (originally reported in Yu et al., 2008 – PMID: 18676989, yet the mechanism is also unknown). Here, we have first cloned ITSN1 WT and ITSN1 W949E+Y965E mutant in lentivirus so they can be introduced and experimentally tested in the ITSN1 KO chromaffin cells (rescue experiments). We found that ITSN1 can rescue exocytic defects of ITSN1 KO cells in terms of capacitance recordings and amperometry, and that the endophilin1-ITSN1 interaction is responsible for the altered secretion phenotype in the chromaffin cells without ITSN1. These results nicely complement our findings that endophilin1-ITSN1 interaction underlies exocytic defects in the endophilin-deficient chromaffin cells.
- 5) We have now included WT or littermate controls in all electrophysiological and electrochemistry experiments. Specifically, we have included WT controls, in addition to the littermate controls, in the electrophysiological and electrochemical experiments in the new Figure 1 and new Suppl. Fig. S1: these data demonstrate that there is no difference between WT and control littermate (endophilin 1KO-2WT-3KO) chromaffin cell secretion, likely due to a protein redundancy (endophilin 1 and 2

seem to have redundant functions in exocytosis, e.g. Figure 2). Following the norms in the chromaffin cell physiology field, for all other electrophysiological and electrochemical experiments (new Figures 2, 6, 7, 8, S2 and S6), either the littermate control, or endophilin TKO rescued with the re-introduced endophilin expression, are added.

Minor concerns:

1. Fig. 1E and F, statistics showed similar colocalization level of CgA with EndoA1 and EndoA2, however, overlay images in C and D don't match very well (more red spots in C WT, only yellow and green spots in D WT). But this is a rather minor drawback.

Colocalization analysis of endogenous CgA/endophilin 1 and CgA/endophilin 2 signals (detailed in Methods) revealed comparable data. Perceived differences in the overlay images originate from individual scaling of each cell, which were not adjusted adequately in the presented example. Note that CgA is a LDCV cargo protein and endophilin 1 and 2 are cytosolic/membrane-associated proteins with many interactors and functions in the cell, thus a complete overlap of signals cannot be expected.

2. In Fig. 3H, TKO and TKO-endoA2 spike frequency do not seem to be different, likely because the trace covers a large time frame. Please add an inset plotting the trace in larger time scale showing the differences in the spike frequency between the two traces. In addition, it's not clear the total analyzed duration of the trace in bar graph Fig. 3J-Q. Please clarify.

We have added the inset in the new Figure 2H. We analyzed 210 seconds-long recordings for each cell: this is now indicated in the figure legend.

3. Scale bar in Fig. 4A is 100 nm, which is incorrect.

Thank you for noticing this error: the correct scale bar size has been indicated in the new Suppl. Fig S3.

4. Fig. 5 and 6 were not the major findings of this study, which can be presented in supplementary figures. In Fig.6E, the bright field image of TKO is much larger than the m-CLING image. It seems that they are not from the same focal plane.

We agree that data presented in Figures 5 and 6 are not the major findings of the study, yet they are essential for the study's conclusions. It was important to carefully examine the exocytic and endocytic machinery in endophilin TKO cells. We consider that it is useful to show some of these data in the main figures (much additional data are presented in the Suppl. Figures S4 and S5) since supplementary figures often do not receive as much attention as the main figures. Notably, some datasets carry vital messages, e.g. no changes in the number of CgA-labelled LDCVs in the whole cell, no changes in the synaptotagmin-1 protein levels (exemplary LDCV-resident protein) on the LDCVs in endophilin TKO cells, no

major alterations in the level and distribution of endocytic proteins in endophilin TKO cells – the latter data are surprising findings in our view, etc.

The bright-field image from each cell was acquired before recording the mCLING fluorescent signals in the red channel. The focal plane is sometimes slightly different in order to acquire optimal signals from the mCLING channel. This is now noted in the figure legend of new Figure 5.

5. In top panel of Fig.8A, it seems that the images were not from the same focus. The nucleus contains white spots in the resting condition, but no white spots in high K⁺.

The examination of distribution of endogenous ITSN1 and ITSN2 proteins upon stimulation (originally Fig. 8A-D, now Fig. 7A-D in the revised version) was done on many cells, and each panel (A or C) contains four different cells depicted in the central cross section. Thus, the cells shown in resting and the high K⁺ conditions are indeed different cells from coverslips treated respectively before fixation and subsequent immunostaining. This is now also stated in the figure legend. That said, we see fast re-distribution of overexpressed ITSN1-EGFP to the plasma membrane proximity upon stimulation in mouse chromaffin cells: this is shown in the new Movie 5.

Reviewers' Comments:

Reviewer #3:

None

Of note, all reviewers comments were addressed. REVIEWERS' COMMENTS: Reviewer 3 I suggest publication of this manuscript. Thank you for the final round of processing of this manuscript.